# Coverage bias in small molecule machine learning

**Fleming Kretschmer** [1], **Jan Seipp**[2], **Marcus Ludwig** [1,3], **Gunnar W. Klau** [2] & **Sebastian Böcker** [1] ✉

Small molecule machine learning aims to predict chemical, biochemical, or biological properties from molecular structures, with applications such as toxicity prediction, ligand binding, and pharmacokinetics. A recent trend is developing end-to-end models that avoid explicit domain knowledge. These models assume no coverage bias in training and evaluation data, meaning the data are representative of the true distribution. However, the domain of applicability is rarely considered in such models. Here, we investigate how well large-scale datasets cover the space of known biomolecular structures. For doing so, we propose a distance measure based on solving the Maximum Common Edge Subgraph (MCES) problem, which aligns well with chemical similarity. Although this method is computationally hard, we introduce an efficient approach combining Integer Linear Programming and heuristic bounds. Our findings reveal that many widely-used datasets lack uniform coverage of biomolecular structures, limiting the predictive power of models trained on them. We propose two additional methods to assess whether training datasets diverge from known molecular distributions, potentially guiding future dataset creation to improve model performance.

Machine learning has been successfully used in biochemistry and chemistry for decades. We consider the task of predicting chemical, biochemical or biological properties of small molecules of biological interest from their molecular structure. A recent trend is to develop end-to-end models that avoid the explicit integration of domain knowledge via inductive bias[1]. Noteworthy examples are generative models for novel antibiotics[2] and highly toxic small molecules[3], or classifiers for antibiotic activity[4,5], olfactory perception[6] and enzyme-substrate prediction[7]. The MoleculeNet paper from 2018 presents 17 medium- to large-scale datasets for molecular property prediction[8]. These data are frequently used in machine learning to train and evaluate new models such as graph neural networks and graphormers; notably, the paper has received more than 2000 citations in 5 years.

The fact that one should not use a model outside of its domain of applicability, has been well-known in the chemometrics community. The situation may be compared to spatial bias, where one uses test

(and training) data from a certain geographic location, but makes claims about a model's performance for other geographic location as well[9]. Yet, this problem is usually ignored when training large-scale end-to-end models for predicting molecular properties. Other models[10,11] are pre-trained on larger structure datasets; yet, this cannot get around the distribution bias in training and evaluation data for individual molecular properties. Recently, words of warnings have emerged that machine learning may result in a reproducibility crisis in science[9,12]. In particular, the datasets from MoleculeNet have been criticized[13,14]. Whereas it is comparatively simple to train a machine learning model that performs well in evaluations, it is much harder to derive a model that indeed contributes to solving the underlying question.

The problem of generalization within a dataset has been extensively researched. For small molecules, the widely-used scaffold split ensures that evaluation is performed for scaffolds not seen in the

[1]Chair for Bioinformatics, Institute for Computer Science, Friedrich Schiller University Jena, Jena, Germany. [2]Algorithmic Bioinformatics, Institute for Computer Science, Heinrich Heine University Düsseldorf, Düsseldorf, Germany. [3]Currently at Bright Giant, Jena, Germany. ✉e-mail: sebastian.boecker@uni-jena.de

training data. This allows us to asses a model's ability to extrapolate to novel molecular structures. However, doing so does not account for the differences in distribution of a molecular property[13]. In particular, small structural changes can entail large differences in the associated molecular property to be predicted, a phenomenon known as "activity cliff"[15]. Finally, all conclusions drawn on the extrapolation performance by employing a scaffold split still only apply to the dataset itself and its own (restricted) chemical space. A model trained and evaluated on a dataset solely consisting of lipids may generalize well between different lipid classes, but there is no reason to assume that it also works for flavonoids.

To ensure that a model is not used outside of its domain of applicability, we have to make sure that the data available for training and evaluating the model are a representative subset of the space of all molecules of interest. It is understood that a dataset that is not sufficiently comprehensive will not allow us to learn all aspects of the problem approached. Yet, even for datasets that contain many thousands of samples, the choice of small molecules included in any such dataset is often far from random. For datasets that rely on experimental measurements, it is instead governed by availability of compounds and, hence, often by monetary aspects. The availability of a compound depends on aspects such as difficulty of chemical synthesis, commercial availability of precursor compounds, and similar considerations from synthetic chemistry and biotechnology. The lower the availability of a compound, the higher the price, and the less likely this compound can be found in large-scale datasets. Clearly, this introduces bias into the training data. For datasets with experimental measurements, unavailability of certain compounds is not going to change in the near future.

To consider the training data distribution for small molecules necessitates some way of estimating the similarity or dissimilarity between molecular structures. Unfortunately, this is a highly intricate problem, and is currently being approached in two ways with individual shortcomings: Firstly, molecular fingerprints allow for a swift processing of large datasets[16]. Yet, measures based on molecular fingerprints are known to exhibit undesirable characteristics[17–25]. In particular, measured distances may differ substantially from chemical intuition. Second, methods based on computing the Maximum Common (Edge) Subgraph better capture the chemical intuition of structural similarity[26–28], see below, but unfortunately, require to solve computationally hard problems.

Here, we show how to inspect a molecular structure dataset for its coverage of small molecule structures of biological interest ("biomolecular structures" for the sake of brevity). Our approach combines Uniform Manifold Approximation and Projection (UMAP) embeddings[29] and computation of structural distance via the Maximum Common Edge Subgraph (MCES). We introduce the *myopic MCES distance* (mMCES distance), which is the informative exact MCES distance for closely related molecules or a good approximation thereof, in case an exact computation is not required and too costly. We demonstrate how we can compute this distance swiftly in practice using a combination of fast lower bounds and integer linear programming. We then show that the distribution of compound classes, as well as a measure for natural product-likeness[30,31] can give good indications on whether the distribution of molecular structures in a dataset differs substantially from that of biomolecular structures. Finally, we shortly discuss shortcomings of the well-known Tanimoto coefficient for performing this type of analysis.

## Results
### Distribution of biomolecular structures
It is understood that we do not know the true "universe of small molecules of biological interest", as this includes small molecules yet to be discovered[32]. Here, we use a combination of 14 molecular

structure databases (*biomolecular structures* for short) as a proxy of this space. These databases contain metabolites, drugs, toxins and other small molecules of biological interest. As used here, the union of databases contains 718,097 biomolecular structures, see the Methods section and Supplementary Table 1 for details. Clearly, this proxy is and will be incomplete; yet, restrictions on the domain of applicability may already be visible against this proxy.

Given a pair of molecular structures, we computed a distance using their Maximum Common Edge Subgraph. To speed-up computations, we estimated (provably correct) lower bounds of all distances. We performed exact computations only if the distance bound is at most a chosen distance threshold, which we set to 10, unless stated otherwise. If the lower bound was above the threshold, we used this bound instead as a distance estimate. Similarly, if the exact distance was computed and above the threshold, we used the threshold instead. We used UMAP[29] to visualize the universe of biomolecular structures in a 2-dimensional plot.

To avoid both proliferating running times and cluttered plots, we uniformly subsampled 20,000 biomolecular structures (Fig. 1). Total running time for MCES computations was about 15.5 days on a 40 core processor. To monitor the effect of subsampling, we uniformly subsubsampled nine times 10,000 molecular structures from this set. We present corresponding UMAP embeddings in Fig. 2, to reveal variations. We observe that subsampling may indeed change the general layout of the UMAP embedding, but that the general layout is often surprisingly similar. It is well-known that these UMAP embeddings have to be interpreted with care[33], see also below.

Certain molecular structures and compound classes, in particular certain lipid classes, result in outlier clusters in the UMAP embedding (Fig. 1). To avoid that these molecular structures dominate the UMAP embedding, we excluded them. This leaves us with 18,096 molecular structures, which will be considered in all following analyses. See Fig. 3 for the resulting map of biomolecular structures, where we have color-coded compound classes according to ClassyFire[34]. Excluded molecular structures can nevertheless be displayed in the UMAP embedding (Supplementary Fig. 1).

Above, we used UMAP to visualize MCES distances. Clearly, any other method for projecting high-dimensional data given as distances (for instance, t-SNE[35], multidimensional scaling[36] or Minimum Spanning Trees) can also be applied (Supplementary Fig. 2)[37–39].

Figure 4 shows the distribution of myopic MCES distances. As expected, most distances are large; yet, for every molecular structure, the smallest distance to another molecular structure is usually below 10 (Fig. 4b). In fact, we observe that distance 10 occurs much more often than what we would expect by chance. This is due to the threshold $T = 10$ used in our computations and, in particular, due to the double thresholding: In case the exact MCES distance is computed but turns out to be larger then $T$, we instead use $T$ as the myopic distance. Given that there is no such hump in the distribution of myopic MCES distances (Fig. 4a), we argue that this thresholding artifact has little effect on the computed UMAP embeddings.

Chari et al.[33] showed that one must be highly cautious deducing structure of the data from a 2-dimensional UMAP embedding. We stress that this is not a restriction for what we are doing: We already know that our data are structured, in the sense that molecular structures can be (dis)similar, or belong or not belong to the same compound classes. What we are investigating is to what extent a subset of the data is a uniform subsample: If we are able to spot non-uniformness in the 2-dimensional UMAP embedding, then it is presumably not a uniform subsample in higher dimensions, either. It is nevertheless indisputable that the UMAP embedding is far from perfect, meaning that a small/large distance in the plot does not necessarily imply a small/large MCES distance (Supplementary Fig. 3). UMAP shares these restrictions with any method that projects a high-dimensional space

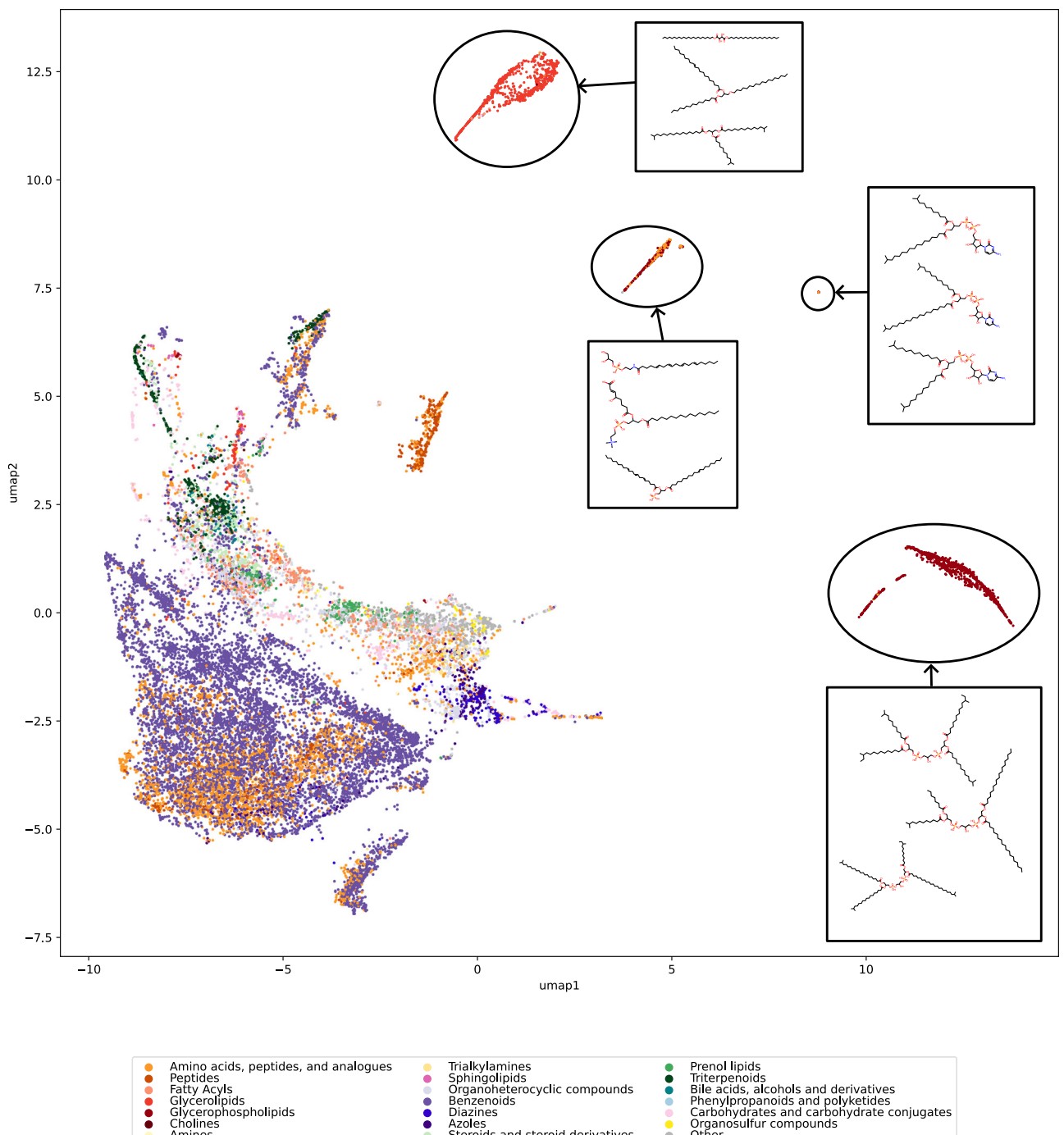

**Fig. 1 | Initial map of biomolecular structures.** Outlier clusters are highlighted and annotated with three exemplary structures drawn at random for each cluster. Different from Fig. 3 and Supplementary Fig. 1, the UMAP embedding was computed from all 19,994 subsampled biomolecular structures. This includes the 1898 molecular structures removed from Fig. 3. In contrast, Supplementary Fig. 1 shows the 1898 molecular structures added back into the UMAP embedding computed from 18,096 molecular structures. Compound classes were chosen based on frequency in the biomolecular structures. For compounds belonging to more than one compound class, the class with the largest structural pattern is selected[34]. Source data are provided as a Source Data file.

into the plane. Clearly, there is also some arbitrariness in the layout of the UMAP plot, see ref. 40 and Fig. 2. Finally, when inserting new samples into an existing UMAP embedding, those new samples tend to be inserted into the existing structure of the plot, rather than generating novel distant clusters or singletons. Compare Fig. 1 and Supplementary Fig. 1: Whereas the lipid classes form distant clusters in the original UMAP embedding, they are integrated into the existing plot structure when reinserted.

## Distribution of molecular structures in public datasets

We consider ten public molecular structure datasets frequently used to train machine learning models[41–53]. See Supplementary Table 2 and the "Methods" section for details. All of these datasets have in common that molecular structures are labeled by experimentally determined measures. Also, all of these datasets have repeatedly been used to train and evaluate machine learning models[8,54,55]. Not all molecular structures in a dataset are biomolecules: See the discussion of natural

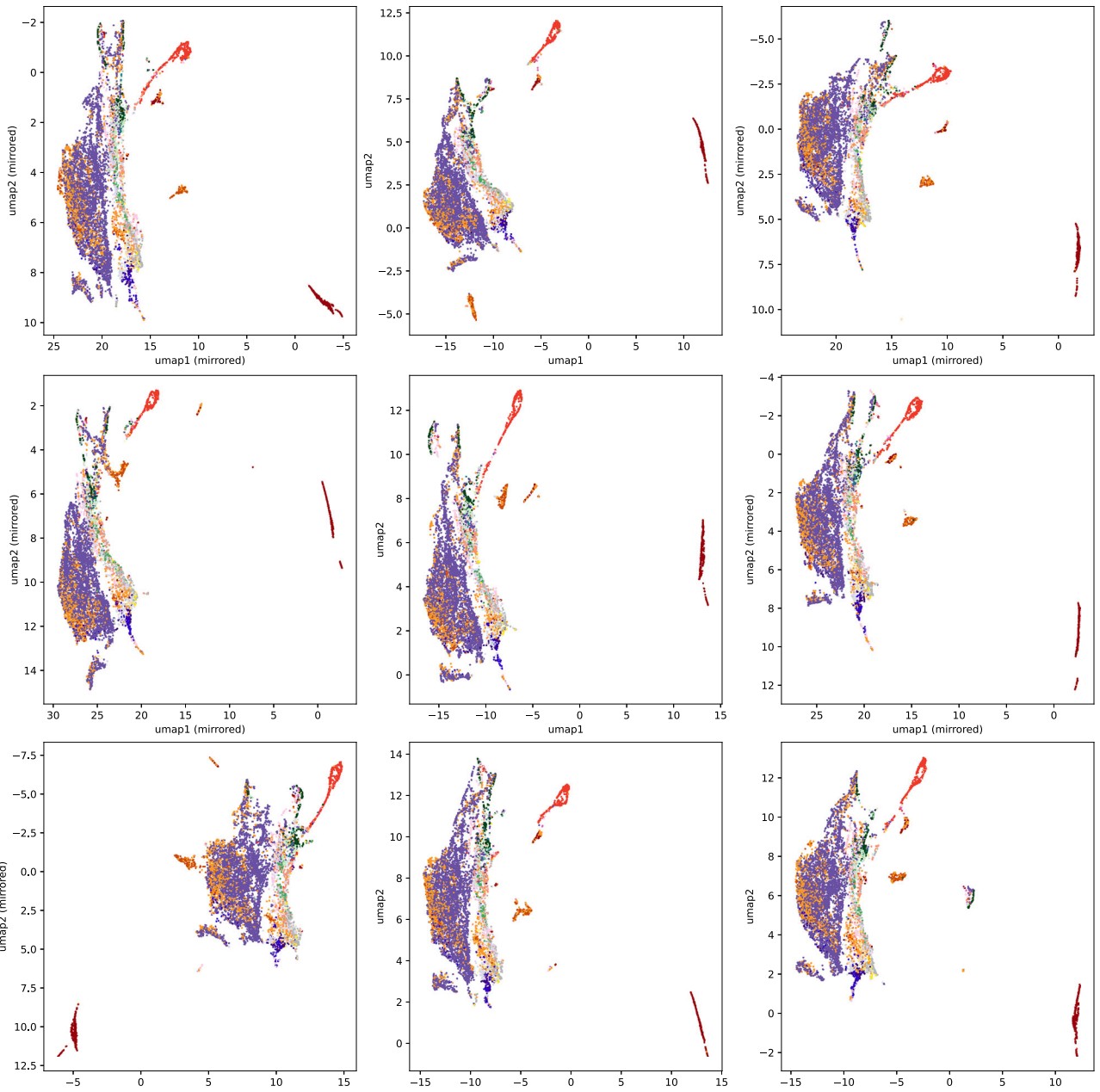

**Fig. 2 | UMAP embeddings vary when other biomolecular structures are subsampled.** It is understood that the (random and uniform) choice of 19,994 biomolecular structures affects the UMAP embedding. To investigate the effect of subsampling, we further uniformly subsubsampled nine times 10,000 structures, and created the corresponding UMAP embeddings. Compound classes are color-coded as in Fig. 1. For easier visual inspection, some of the plots have been mirrored, as indicated. Source data are provided as a Source Data file.

product-likeness score distributions below. Even less so, all molecular structures in a dataset are contained in the 20,000 subsampled biomolecular structures, see again Supplementary Table 2 for the statistics.

We investigated to what extent the molecular structures in each dataset are a uniform subset of biomolecular structures. For each dataset, we computed myopic MCES distances for all molecular structures, combining the training dataset and the 18,096 biomolecular structures. Nine resulting plots can be found in Fig. 5 and one in Supplementary Fig. 4. To ensure comparability of plots, we used the same UMAP embedding as in Fig. 3. We observe that the subset of molecular structures available in public datasets is usually far from uniform. We stress that we can make this observation based on the

available UMAP embeddings, without the need that these plots represent some chemical or biochemical truth. We argue that most of the public datasets are also not representative, meaning that large areas of the biomolecular structures are completely missing in the datasets. In fact, some datasets are concentrated in one or few areas in the plot.

Recall that additional samples tend to get inserted in the existing structure of the plot even if they are different from all existing samples (Fig. 1 and Supplementary Fig. 1). Hence, even molecular structures highly different from all biomolecular structures will not result in novel outlier clusters. Arguably the best coverage is observed for the toxicity datasets and the MS/MS dataset. Given the known limitations of the UMAP plot, this does not imply that these datasets contain a uniform

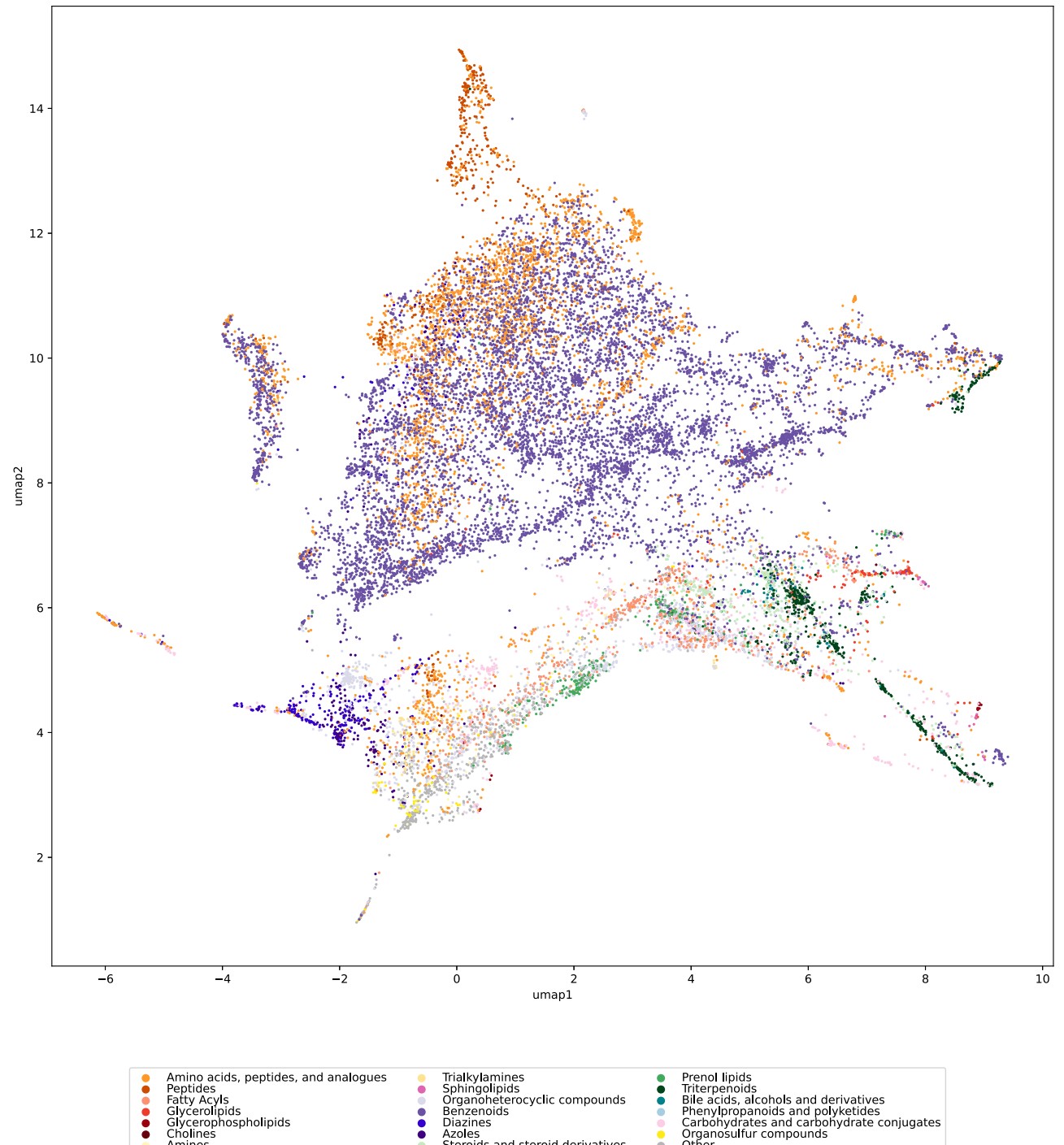

**Fig. 3 | Map of biomolecular structures with outlier clusters removed.** Certain compounds were excluded before computing the UMAP embedding and the plot, compare to Fig. 1. The UMAP embedding was computed from the remaining 18,096 molecular structures. See Supplementary Fig. 1 for the corresponding plot where these compounds were reinserted. Compound classes are color-coded as in Fig. 1. Source data are provided as a Source Data file.

or representative subset of molecular structures. Instead, these plots can only warn us that a dataset is not representative.

When evaluating machine learning models, molecular structures are often split according to scaffold structures. This allows us to estimate performance measures more indicative of generalization ability, than uniformly splitting the dataset into train and test partitions[13]. We computed scaffold splits for the nine training datasets from Fig. 5 using Bemis-Murcko scaffolds[56] from DeepChem[57] (Supplementary Fig. 5). Next, we computed a ten-fold scaffold split for the dataset BBBP

(Supplementary Fig. 6). Scaffold splitting results in a non-uniform distribution of train and test data; this is particularly the case for the first two folds of the ten-fold scaffold split of BBBP (Supplementary Fig. 6). Yet, even scaffold splits can be misleading with respect to a model's generalization performance: Whereas large subregions of biomolecular structures have no coverage from the molecular structures in the training data, we observe that almost all molecular structures in the test split have a somewhat close molecular structure in the train split.

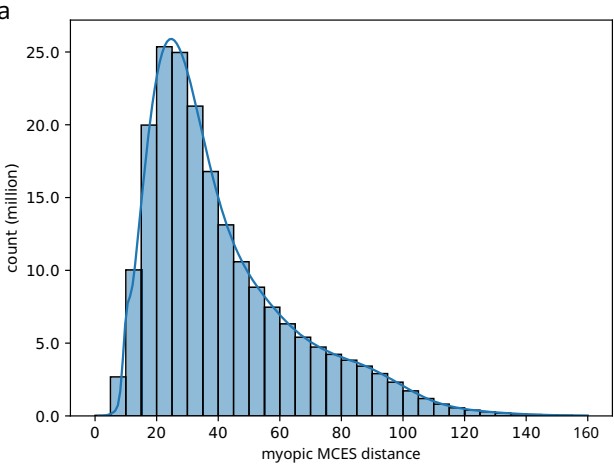

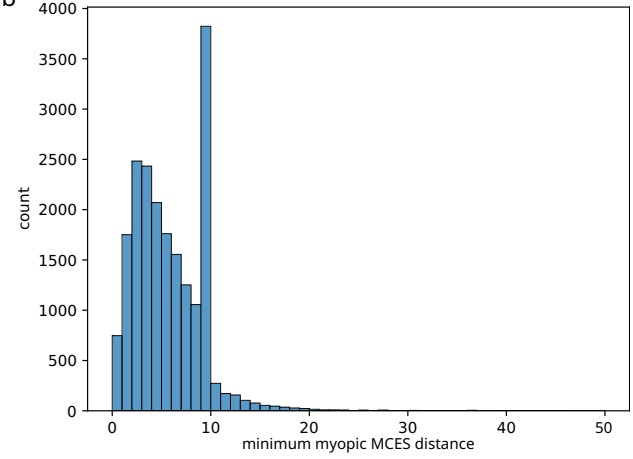

**Fig. 4 | Distribution of Maximum Common Edge Subgraph distances.** Myopic MCES distances were computed with threshold $T = 10$. For the histogram plots, bin intervals are left-open and right-closed. For example, the tenth bin in the right plot covers distances $x$ with $9 < x \leq 10$. **a** The distribution of all 199,870,021 computed distances from the subsampled biomolecular structures are shown as a histogram plot and kernel density estimate. The plot is truncated after distance 160. **b** Histogram of the minimum myopic MCES distance for each subsampled biomolecular structure against all other biomolecular structures, truncated after distance 50. The hump is due to the employed threshold $T = 10$, see text for details. Source data are provided as a Source Data file.

## Maximum Common Edge Subgraph computations

The Maximum Common Edge Subgraph problem is well-suited to capture high similarities between molecular structures[26–28,58]. The *MCES distance* between two molecular structures equals the number of edges in the first molecular graph plus the number of edges in the second molecular graph, minus twice the number of edges in the MCES. This distance agrees well with chemical intuition: It is the minimum number of edges we have to remove from both graphs so that the resulting graphs are isomorphic, ignoring singleton nodes. From a chemical standpoint, we can think of it as the number of chemical reactions required to transform one molecule into another. Molecular graphs are labeled graphs where nodes are associated with atom types and edges with bond orders. When comparing two such graphs we have to take into account these labels and how to compare them, such as double vs. aromatic bond, see the Methods section for details. We do not consider hydrogen atoms in our computations.

Unfortunately, computing the MCES is provably hard, and it is considered highly unlikely that an efficient algorithm for this problem exists. Precisely speaking, the MCES problem is NP-hard, as it generalizes subgraph isomorphism which, in turn, generalizes the clique problem. As MCES is clearly in NP, it is NP-complete. Next, an "efficient algorithm" is an algorithm with running time polynomial in the size of the instance, that is, the number of edges or nodes of the two graphs for the MCES problem. It is known that there cannot be an algorithm with polynomial time for an NP-hard problem, unless P = NP. The field of theoretical computer science usually assumes that P ≠ NP holds, and there exists a Millennium prize to prove or disprove this. But even if P = NP, it is assumed that the exponent in the running time of any polynomial time algorithm will be prohibitively high, and the resulting algorithms will be of no practical use, similar to the polynomial-time algorithm for PRIMES[59].

To compute distances for our UMAP embedding in Fig. 3 required to swiftly solve more than 160 million instances of the NP-hard MCES problem. We present an efficient implementation based on computing lower distance bounds to quickly recognize dissimilar structure pairs, with an additional step for potentially similar pairs for which we compute provably exact distances with an Integer Linear Programming (ILP) formulation. Using an ILP to compute the MCES has two key advantages over previous methods based on enumerating cliques in a product graph: (i) Contrary to the clique-based methods, the ILP tends

to be fast when the input is similar, which, as we argue, are exactly the interesting cases and (ii) the ILP avoids the cumbersome treatment of the so-called ΔY exchange that occurs when modeling the problem via line graphs. Our method is the first to use an ILP for the comparison of chemical structures.

In practice, we decide for a distance threshold (say, $T = 10$ edge modifications) whether our bounds guarantee that the true distance is at least $T$; in these cases, we use the bound as an approximation of the true distance. Only if no bound can guarantee that the distance is at least $T$, we execute the exact algorithm and report its result, this time using $T$ as an upper bound. We call the resulting distance the *myopic MCES distance*. The two-step procedure has two advantages: (a) The ILP is usually fast if the MCES distance is small, whereas running times can get very large for larger distances. Our two-step approach thus explicitly and efficiently excludes most running time-intensive exact computations. (b) Considering a pair of molecular structures, it is obviously of high interest to know whether the MCES distance between those two structures is 2 or 8. Yet, we argue that it is mostly irrelevant to know whether the MCES distance is 42 or 48: In both cases, the two molecular structures are highly dissimilar. Both numbers can serve as reasonable approximations for the true distance. In particular, divergence in the larger distance will not result in substantial changes of the UMAP embedding, by formulation of the UMAP optimization problem[29].

We performed an in-depth evaluation of our methods using a subset containing 20,000 uniformly subsampled instances, where an *instance* is a pair of biomolecular structures (Fig. 6). Running times were measured on a 40-core processor running 80 threads in parallel; we report running times per thread. For the ILP, 24 of 20,000 instances did not finish within four days of wall clock time (Supplementary Fig. 7). For those instances, we use the time at which computations were stopped as a running time proxy. Doing so, total running time of the ILP equals 234.2 days for the 20,000 instances (average 16.9 min per instance). Sorting instances by ILP running time, we observe that 1.04% of the instances are responsible for more than 95% of the total running time.

We first consider the dependence of ILP running times on the exact MCES distance (Fig. 6a, b). ILP instances that did not finish are excluded from this plot. For distances up to 75, we observe a clear correlation between distance and running time. For example, 29.5% of

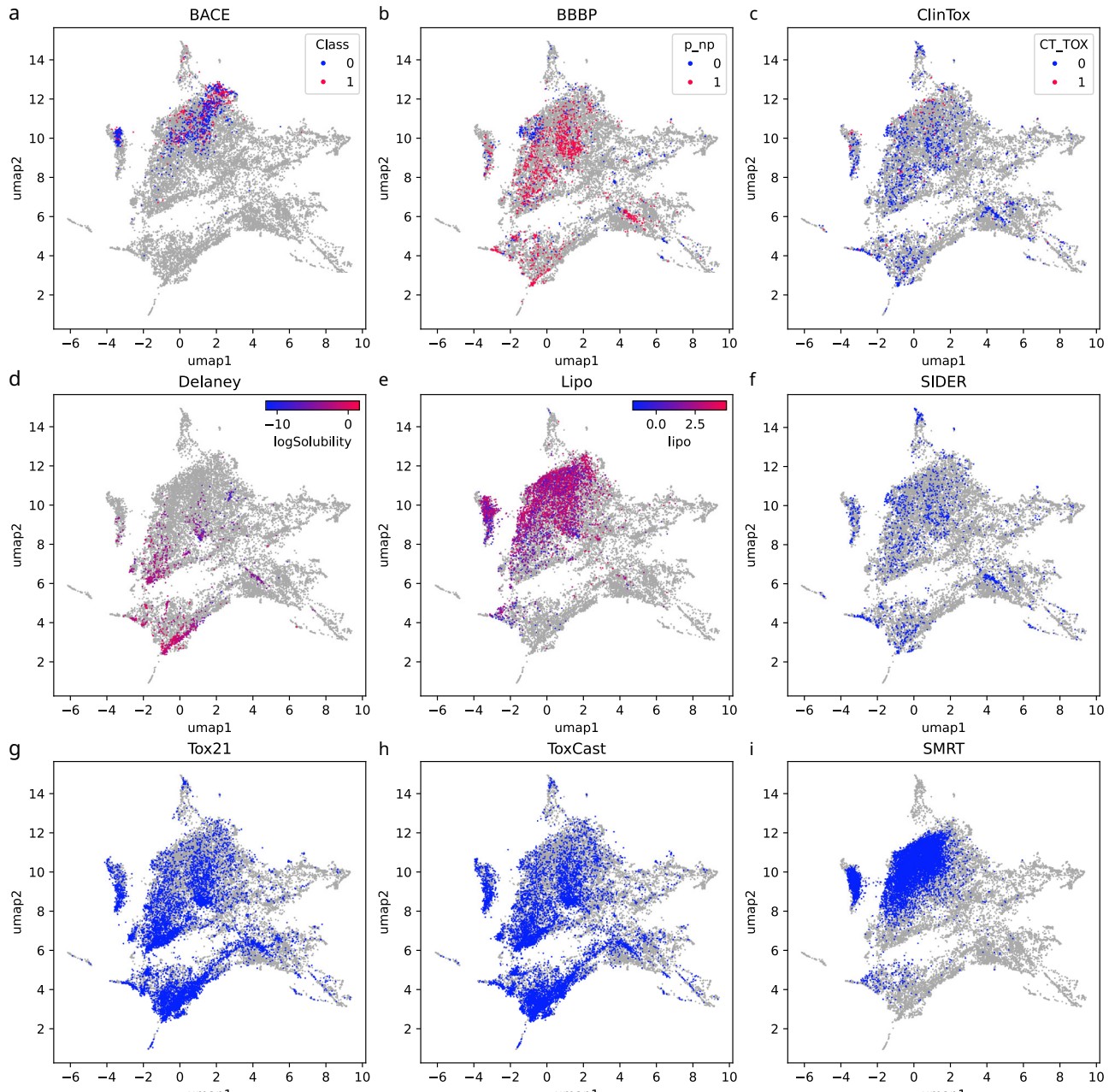

**Fig. 5 | Maps of nine public molecular structure training datasets. a–i** UMAP embedding of nine public datasets that are frequently used to train, evaluate, and compare machine learning models. Shown are datasets BACE (**a**), BBBP (**b**), ClinTox (**c**), Delaney (**d**), Lipo (**e**), SIDER (**f**), Tox21 (**g**), ToxCast (**h**), and SMRT (**i**). We use the same UMAP embedding as in Fig. 3. No molecular structures fall outside of this plot. Number of samples differ for each plot, as each plot combines the 18,096 biomolecular structures and the structures from the dataset. See Supplementary Table 2 for details. Biomolecular structures are shown in light gray. For datasets with single target variables, we color-code these in the plots. See Supplementary Fig. 12 for the corresponding plots using ChEMBL as the background distribution. Source data are provided as a Source Data file.

the instances have MCES distance up to 30 but contribute only 0.91% to the total running time. For larger distances up to 100, this correlation becomes less clear. Apparently, both the MCES distance and the actual structure of the MCES instance are affecting running times. Beyond distance 100, results must be interpreted with care, as there are few such instances corresponding to few outlier structures.

Second, we evaluated how the combination of bounds and exact computations results in favorable running times (Fig. 6c, d). We concentrate on thresholds $10 \leq T \leq 25$ reasonable for measuring molecular structure similarity. Increasing the threshold $T$ means that more instances have to be computed exactly since fewer instances result in a

lower bound above $T$. Compared to the exact method, we observe a massive running time improvement: Even for the largest threshold $T = 25$, total and average running times decrease 1101-fold. Further lowering the threshold also further reduces running times: Using $T = 10$ instead of $T = 25$ decreases the total and average running times 5.3-fold. For the myopic MCES distances, we observe right humps in the distributions of running times (Fig. 6c). We attribute these humps to instances that have to be computed exactly.

To exclude bias through subsampling, we repeated the above analysis using all pairs from the 19,994 biomolecular structures (see Section Subsampling molecular structures). To avoid proliferating

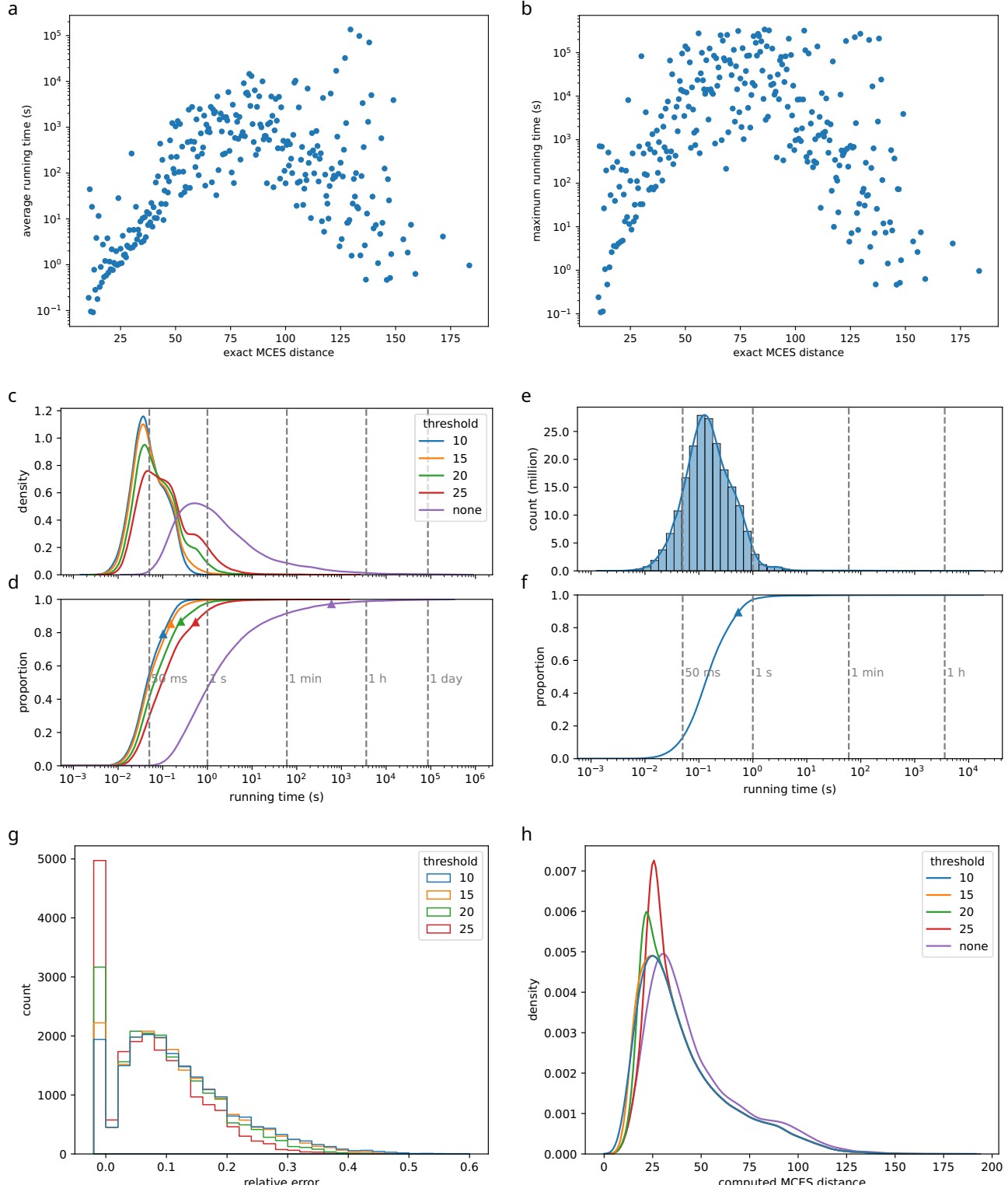

**Fig. 6 | Computing the Maximum Common Edge Subgraph.** Each instance is a pair of molecular structures from the biomolecular structures. For (e,f) we use all 199,870,021 instances from our subsampled biomolecular structures, for all other plots a uniform subsample of 20,000 instances. **a**, **b** Running time of the ILP vs. exact MCES distance of the instance. Shown are (**a**) average and (**b**) maximum running times for each MCES distance bin. Note the logarithmic y-axis. **c**, **d** Distribution of running times using a MCES distance thresholds $T = 10, 15, 20, 25$ and no threshold. Note the logarithmic x-axis. **c** Kernel density estimates and (**d**) total running times (empirical cumulative distributions). Average running times shown as triangles. **e**, **f** Distribution of running times for $T = 10$ using all instances. Note the logarithmic x-axis. The distribution contains outliers not visible in the plot:

213 instances (0.0001%) have a running time longer than 1 h. **e** Histogram and kernel density estimate, (**f**) total running time (empirical cumulative distribution). Average running time shown as a triangle. **g** Using bounds instead of exact computations results in an error of distance estimates. If we increase threshold $T$, then more instances are computed exactly. Shown is the histogram of relative errors for different thresholds, using bin width 0.02 and truncated at 0.6. Intervals are left-open; the first bin constitutes only instances with a (relative) error of exactly zero. **h** Kernel density estimates of myopic and exact MCES distance distributions ("computed MCES distance"). Compare to Fig. 4. Source data are provided as a Source Data file.

running times — here and throughout the rest of the paper — we concentrated on threshold $T = 10$, see Fig. 6e, f. Total running time for the complete set was 1240 days, corresponding to 15.5 days of wall clock time on a 40 core processor. We observe basically the same pattern as for the subsampled instances, but the average running time per instance is 5.2-fold larger than for the subsampled set (536 ms vs. 102 ms). Here, 0.2% of the instances are responsible for half of the total running time. The difference in average running time for the same threshold can likely be attributed to a difference in ILP solver settings, see the Methods section for details. We did not use a running time limit for the ILP solver, see Methods section.

Finally, we analyzed the error of computing bounds instead of exact distances: The myopic MCES distance may differ from the exact distance; the *absolute error* is the difference between the exact distance value and the myopic MCES distance. Recall that the myopic MCES distance is a lower bound, so the absolute error is never negative. Deviations from the true value are relevant mostly in comparison to the true value. To this end, we investigate the *relative error*, dividing the absolute error by the exact distance value (Fig. 6g). As expected, more distance estimates have a non-zero error if we lower threshold $T$: In detail, 22.5% of instances have zero error for $T = 25$; this decreases to 9.5% for $T = 10$. Beyond this, we observe that varying the threshold $T$ between 10 and 25 does not substantially change the shape of the relative error distribution. We observe that the distribution of myopic MCES distances (compare to Fig. 4) is very similar for different thresholds and for exact computations, only shifted (Fig. 6h).

Recently, an implementation of RASCAL[60] has become available in RDKit (RDKit: Open-source cheminformatics, https://www.rdkit.org). RASCAL solves the MCES problem, computing the exact MCES when a predefined similarity threshold is exceeded. Similar to our implementation, the results of computed bounds can be used to approximate similarity when no exact calculations are performed. In contrast to the (absolute) myopic MCES distance, a relative measure of similarity introduced by Johnson[61] is used. To evaluate RDKit's RASCAL implementation, we used the same subset of 20,000 instances as above. Using the default similarity threshold of 0.7, only 280 of these instances (1.40%) were calculated exactly. Notably, 100 of those 280 instances (35.7%) failed to compute, either because an internal limit on the size of the product graph ("maxNumberMatchingBonds") was exceeded (98 instances), or because the time-out of one hour was reached (two instances). Disabling the internal size limit on the product graph, 68 of those 98 instances were computed exactly in less than one hour. In 29 cases, the running time threshold was reached and in one case, the available memory of 256 GB was exceeded. For all 279 exactly calculated instances where the memory was not exceeded, the average wall-clock per instance was 7.92 min, using one hour as proxy for instances with time-out.

Next, to compute more instances exactly, we lowered the similarity threshold to 0.5. Doing so, 2899 (14.5%) of the instances were calculated exactly. Of these, 423 (14.6%) of these calculations did not finish due to the internal size limit (408 instances) or time-outs (15 instances). When disabling the internal size limit, 186 of 408 instances exceeded the time limit of one hour; for five instances, the memory was exceeded. Here, the average wall-clock time for the exactly calculated 2894 instances was 4.47 min, disregarding the instances were the program crashed because memory usage was exceeded. Again, for instances with time-out one hour was used as proxy.

Notably, the internal size limit can only be changed by modifying and re-compiling RDKit's C++ source code; the strict size limit was a deliberate choice of the developers to prevent excessive memory usage. Also, the high memory usage of this approach makes it basically impossible to parallelize computations. Finally, computing approximate results via RDKit's RASCAL is orders of magnitude faster than computing MCES bounds, with a mean wall-clock time of 1.36 ms for the former. Yet, comparing running times between RDKit and our

MCES code is misleading, as it is dominated by running time differences between compiled C++ and interpreted Python code.

Numerous variants of the MCES problem exist, such as finding a Maximum Common Subgraph (MCS), a connected MCES, or restricted variants of the problem[28]. It is extremely challenging to compare the quality of results from different such variants, so we refrain from an in-depth evaluation. Seipp[62] performed running time evaluations of the myopic MCES distance against MCS computations using RDKit ("rdFMCS" module as opposed to "rdRascalMCES", which contains the RASCAL implementation) and SMARTScompare[63] and found that the myopic MCES approach is several orders of magnitude faster than exact MCS methods implemented there, and sometimes even on par with heuristic methods[58]. This is noteworthy as the MCES problem is usually assumed to be substantially harder than the MCS problem.

For numerous computational tasks, it is required that a distance measure is a metric. We can easily show that both the bounds as well as the exact MCES distances are (pseudo)metrics. For the myopic MCES distances, the important fact is that we apply double thresholding: On the one hand, the exact distance is only computed in case the bounds are below threshold $T$. On the other hand, if the exact distance is above $T$, we instead report $T$ as the myopic MCES distance. This implies that whenever we execute exact computations, the reported distance is smaller or equal to $T$; whereas if we only execute bound computations, the reported distance is greater or equal to $T$. Doing so speeds up ILP computations via constraint (8). Beyond that, it allows us to prove that the myopic MCES distance is indeed a metric, see the Methods section. If we do not use the second threshold (that is, we return the exact MCES distance even if it is larger than $T$) then this does not only increase running times of the ILP. In addition, the resulting distance measure is no longer a metric: See Supplementary Fig. 8 for an example where the triangle inequality is violated. Instead of double thresholding, we may use the Floyd-Warshall algorithm[64] for finding all shortest paths in a complete graph, to enforce the triangle inequality.

## Compound class distribution

Besides a uniform subsample, there is another feature a molecular structure dataset should exhibit so that it represents the full space of biomolecular structures: Namely, all compound classes that biomolecules belong to should also be present in the training data. If a dataset completely misses molecular structures from a particular compound class, then a machine learning model trained on the data might show poor predictions for this compound class. The same holds true in case very few samples exist for a particular compound class. This will not be noticeable in evaluations: If we split the dataset into test and training data, the compound class is still absent from the test data. Similarly, if only few examples from a compound class are present, then those will have little or no influence on evaluation statistics. Consequently, we may overestimate the power of the resulting machine-learning models for real-world data.

Historically, compound classes were defined based on, say, biochemical precursors. Unfortunately, class annotations were available only for a small fraction of molecular structures. Recently, certain compound class ontologies were defined purely based on molecular structures[34,65]. Here, we concentrate on the ClassyFire ChemOnt ontology, which contains 4825 classes[34].

Throughout this paper, we concentrated on machine-learning models for biomolecular structures. Consequently, we want to ignore compound classes that contain basically no biomolecular structures: If no or few biomolecules are part of a certain compound class, it is not surprising that a molecular structure dataset will also contain no or few molecular structures for that compound class. To this end, we only consider those classes where at least 5% or 1% of biomolecular structures are part of the class, respectively. See Fig. 7 and Supplementary Fig. 9 for the corresponding statistics. There, we have chosen a somewhat arbitrary threshold of 15 molecular structures ought to be

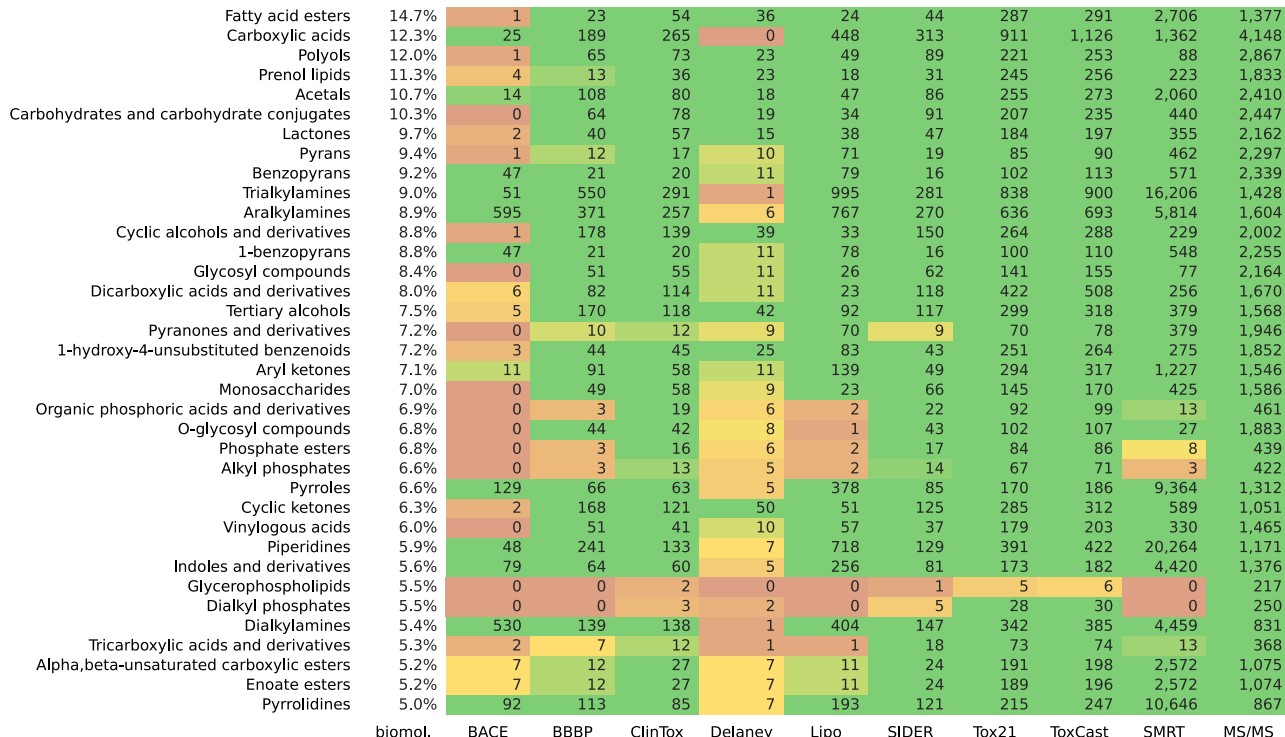

| | biomol. | BACE | BBBP | ClinTox | Delaney | Lipo | SIDER | Tox21 | ToxCast | SMRT | MS/MS |
|---|---|---|---|---|---|---|---|---|---|---|---|
| Fatty acid esters | 14.7% | 1 | 23 | 54 | 36 | 24 | 44 | 287 | 291 | 2,706 | 1,377 |
| Carboxylic acids | 12.3% | 25 | 189 | 265 | 0 | 448 | 313 | 911 | 1,126 | 1,362 | 4,148 |
| Polyols | 12.0% | 1 | 65 | 73 | 23 | 49 | 89 | 221 | 253 | 88 | 2,867 |
| Prenol lipids | 11.3% | 4 | 13 | 36 | 23 | 18 | 31 | 245 | 256 | 223 | 1,833 |
| Acetals | 10.7% | 14 | 108 | 80 | 18 | 47 | 86 | 255 | 273 | 2,060 | 2,410 |
| Carbohydrates and carbohydrate conjugates | 10.3% | 0 | 64 | 78 | 19 | 34 | 91 | 207 | 235 | 440 | 2,447 |
| Lactones | 9.7% | 2 | 40 | 57 | 15 | 38 | 47 | 184 | 197 | 355 | 2,162 |
| Pyrans | 9.4% | 1 | 12 | 17 | 10 | 71 | 19 | 85 | 90 | 462 | 2,297 |
| Benzopyrans | 9.2% | 47 | 21 | 20 | 11 | 79 | 16 | 102 | 113 | 571 | 2,339 |
| Trialkylamines | 9.0% | 51 | 550 | 291 | 1 | 995 | 281 | 838 | 900 | 16,206 | 1,428 |
| Aralkylamines | 8.9% | 595 | 371 | 257 | 6 | 767 | 270 | 636 | 693 | 5,814 | 1,604 |
| Cyclic alcohols and derivatives | 8.8% | 1 | 178 | 139 | 39 | 33 | 150 | 264 | 288 | 229 | 2,002 |
| 1-benzopyrans | 8.8% | 47 | 21 | 20 | 11 | 78 | 16 | 100 | 110 | 548 | 2,255 |
| Glycosyl compounds | 8.4% | 0 | 51 | 55 | 11 | 26 | 62 | 141 | 155 | 77 | 2,164 |
| Dicarboxylic acids and derivatives | 8.0% | 6 | 82 | 114 | 11 | 23 | 118 | 422 | 508 | 256 | 1,670 |
| Tertiary alcohols | 7.5% | 5 | 170 | 118 | 42 | 92 | 117 | 299 | 318 | 379 | 1,568 |
| Pyranones and derivatives | 7.2% | 0 | 10 | 12 | 9 | 70 | 9 | 70 | 78 | 379 | 1,946 |
| 1-hydroxy-4-unsubstituted benzenoids | 7.2% | 3 | 44 | 45 | 25 | 83 | 43 | 251 | 264 | 275 | 1,852 |
| Aryl ketones | 7.1% | 11 | 91 | 58 | 11 | 139 | 49 | 294 | 317 | 1,227 | 1,546 |
| Monosaccharides | 7.0% | 0 | 49 | 58 | 9 | 23 | 66 | 145 | 170 | 425 | 1,586 |
| Organic phosphoric acids and derivatives | 6.9% | 0 | 3 | 19 | 6 | 2 | 22 | 92 | 99 | 13 | 461 |
| O-glycosyl compounds | 6.8% | 0 | 44 | 42 | 8 | 1 | 43 | 102 | 107 | 27 | 1,883 |
| Phosphate esters | 6.8% | 0 | 3 | 16 | 6 | 2 | 17 | 84 | 86 | 8 | 439 |
| Alkyl phosphates | 6.6% | 0 | 3 | 13 | 5 | 2 | 14 | 67 | 71 | 3 | 422 |
| Pyrroles | 6.6% | 129 | 66 | 63 | 5 | 378 | 85 | 170 | 186 | 9,364 | 1,312 |
| Cyclic ketones | 6.3% | 2 | 168 | 121 | 50 | 51 | 125 | 285 | 312 | 589 | 1,051 |
| Vinylogous acids | 6.0% | 0 | 51 | 41 | 10 | 57 | 37 | 179 | 203 | 330 | 1,465 |
| Piperidines | 5.9% | 48 | 241 | 133 | 7 | 718 | 129 | 391 | 422 | 20,264 | 1,171 |
| Indoles and derivatives | 5.6% | 79 | 64 | 60 | 5 | 256 | 81 | 173 | 182 | 4,420 | 1,376 |
| Glycerophospholipids | 5.5% | 0 | 0 | 2 | 0 | 0 | 1 | 5 | 6 | 0 | 217 |
| Dialkyl phosphates | 5.5% | 0 | 0 | 3 | 2 | 0 | 5 | 28 | 30 | 0 | 250 |
| Dialkylamines | 5.4% | 530 | 139 | 138 | 1 | 404 | 147 | 342 | 385 | 4,459 | 831 |
| Tricarboxylic acids and derivatives | 5.3% | 2 | 7 | 12 | 1 | 1 | 18 | 73 | 74 | 13 | 368 |
| Alpha,beta-unsaturated carboxylic esters | 5.2% | 7 | 12 | 27 | 7 | 11 | 24 | 191 | 198 | 2,572 | 1,075 |
| Enoate esters | 5.2% | 7 | 12 | 27 | 7 | 11 | 24 | 189 | 196 | 2,572 | 1,074 |
| Pyrrolidines | 5.0% | 92 | 113 | 85 | 7 | 193 | 121 | 215 | 247 | 10,646 | 867 |

**Fig. 7 | Compound class distribution.** Compound classes are not represented equally in the datasets. Here, all ClassyFire compound classes occurring in at least 5% of the biomolecular structures are investigated. The occurrences for structures of the datasets BACE, BBBP, ClinTox, Delaney, Lipo, SIDER, Tox21, ToxCast, SMRT, and MS/MS are shown. Some molecular structures could not be classified by ClassyFire and were discarded: This applies to seven structures from BBBP, one from SIDER, four from ToxCast, and 1275 biomolecular structures. Compound classes with at least 15 occurrences in all datasets are omitted; this applies to 65 compound classes. See Supplementary Fig. 9 for a larger subset of ClassyFire classes at 1% cutoff.

present for any compound class. We argue that trends will stay the same, independent of the chosen threshold. We only display those compound classes in the two figures for which at least one dataset contains less than 15 examples. For 65 compound classes (threshold 5%) and 95 compound classes (threshold 1%), respectively, none of the datasets contains less than 15 examples. Yet, recall that ClassyFire ChemOnt is an ontology and that a molecular structure may belong to several compound classes simultaneously. Compound classes on higher levels of the ontology can be huge (for example, virtually all molecular structures belong to the class *organic compounds*) and consequently, somewhat uninformative. We provide the full statistics in Supplementary Data 1.

Considering the compound class distribution in a dataset overlaps to a certain extent with MCES UMAP embeddings, compare to Fig. 3. Yet, both approaches also complement each other: The UMAP embedding introduces a certain amount of arbitrariness and may show structure even if there is none[33] (Supplementary Fig. 3). In contrast, compound classes have a biochemical meaning, and we only super-impose this known structure onto the dataset. On the other hand, this restricts our analysis: Compound classes represent true structure in the space of molecules, but by no means, they represent all structure in this space. Hence, compound class analysis cannot detect all short-comings of our training data. In contrast, the UMAP embedding allows to spot issues without the corset of compound classes.

### Natural product-likeness score distributions
As a third approach to test whether a molecular structure dataset reproduces the universe of biomolecules, we propose to study the distribution of natural product-likeness scores. These scores provide a measure of how molecules are similar to the structural space covered by natural products[30]. Natural products, in turn, are chemical entities produced by living organisms. Hence, natural product-likeness scores

allow us to differentiate between biomolecules and synthetic mole-cules presumably not of biological interest. Here, we use the score of Ertl et al.[30] as implemented in RDKit. Notably, we do not claim that this score is a perfect classifier for a compound being a natural product; yet, this is not necessary for spotting major changes in distribution.

Our reason to study natural product-likeness scores is linked to the intrinsic problem of providing a molecular structure dataset with experimental data: As discussed above, the choice of small molecules included in a dataset is usually governed by monetary aspects. Next, the prize of a compound depends on the difficulty of chemical synthesis etc, not its importance for building a machine learning model. Certain molecules may be regarded as highly interesting for a certain dataset, but if those molecules are too expensive, they cannot be measured.

We found that natural product-likeness scores allow us to get an impression how far a particular dataset deviates from the universe of biomolecules. In particular, we can compare the score distribution to that of PubChem[66]: On the one hand, PubChem contains the vast majority of biomolecular structures used here. On the other hand, PubChem additionally contains many million molecular structures that are presumably molecules of no particular biological interest. Hence, a distribution somewhat similar to that of biomolecular structures hints at a dataset where indeed, molecules of biological interest are present in sufficient numbers. In contrast, a distribution similar to or beyond that of PubChem indicates that the dataset contains only a small fraction of biomolecules. If we apply a model trained on this data to real-world data that, according to our assumptions, was measured from a biological source, then the model may again perform much worse than what we would expect from evaluation results. See Fig. 8 for natural product-likeness score distributions of the ten datasets. We observe that for three datasets (BACE, Lipo, SMRT) the distribution of natural product-likeness scores is substantially different from that of

a

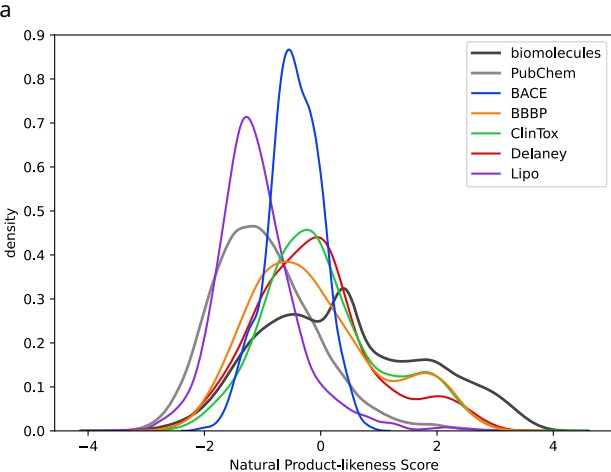

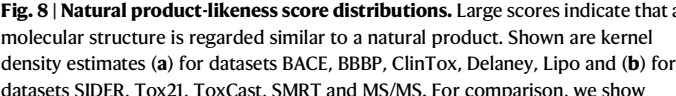

b

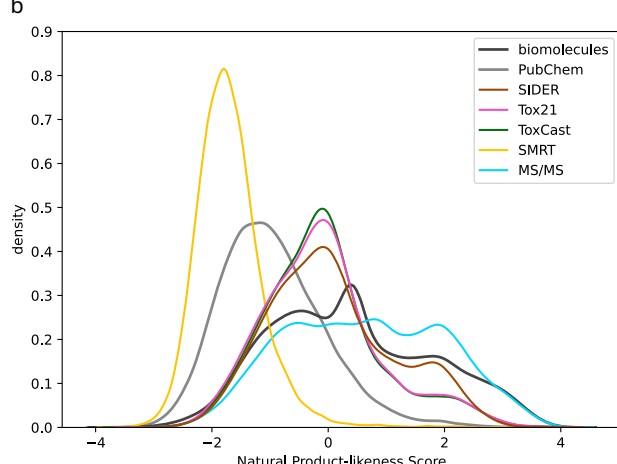

**Fig. 8 | Natural product-likeness score distributions.** Large scores indicate that a molecular structure is regarded similar to a natural product. Shown are kernel density estimates (**a**) for datasets BACE, BBBP, ClinTox, Delaney, Lipo and (**b**) for datasets SIDER, Tox21, ToxCast, SMRT and MS/MS. For comparison, we show kernel density estimates for the 19,994 biomolecular structures and PubChem (20k uniformly subsampled molecular structures). Source data are provided as a Source Data file.

biomolecules. For the other seven datasets, we do not observe a particularly noteworthy distribution pattern.

## Tanimoto coefficients

Tanimoto coefficients[67] are arguably the most commonly employed measure to estimate the (dis)similarity of two molecular structures, and are the working horse for numerous cheminformatics applications including virtual screening. To compute a Tanimoto coefficient, we independently transform each molecular structure into a binary vector, where each position encodes for the presence or absence of a particular subgroup. Then, we compare the two binary vectors using the Jaccard index (Tanimoto coefficient) or the Jaccard distance (one minus Tanimoto coefficient) for similarities and dissimilarities, respectively. One of the main advantages of Tanimoto coefficients is that after transforming each molecular structure into a binary vector, subsequent operations can be executed extremely fast. Consequently, it is most likely that Tanimoto coefficients will remain in use for any application that requires swift computations.

Unfortunately, Tanimoto coefficients also have a number of issues, *because* we are transforming a molecular structure to a binary vector. It is inevitable that substantial information is lost in this transformation. Citing Bajusz et al.[25], "despite the generally positive findings about the applicability of the Tanimoto coefficient several of its weaknesses have also been reported from as early as in a 1998 study." Numerous resulting issues were first discussed by Flower[18], but see also refs. [17,19–25]. For example, similarity values when comparing large structures, for which many bits of the bit vector are Ones, behave very differently from similarity values when comparing small structures, for which almost all bits are Zeros[17,19,23]. These issues hold for any type of molecular fingerprint, be it a predefined list of molecular properties (say, MACCS[68] or PubChem CACTVS[69]) or combinatorial fingerprints (Morgan[16], Extended Connectivity[70]). Notably, these issues have nothing to do with hashing Extended Connectivity fingerprints to, say, 2048 bits[70]. The expected value of the Tanimoto coefficients between two distant molecular structures varies between fingerprint types[20], and is seemingly smallest for MAP4 fingerprints (MinHashed atom-pair fingerprint up to a diameter of four bonds)[71]. Yet, this must not be misinterpreted as a measure of quality of a fingerprint type: If we replace each Tanimoto value by its square, then values become smaller but not more informative. Finally, similar issues hold for *any* similarity measure based on molecular fingerprints

(say, the Sørensen-Dice coefficient), as the encoding into a binary vector is responsible for the issues[18].

In application, Tanimoto coefficients often result in counterintuitive values both for highly similar and highly dissimilar molecular structures: See Fig. 9 for a collection of molecular structure pairs that demonstrate this problem. Unlike machine learning task such as structure-activity or structure-property prediction[72,73], concatenation of molecular fingerprints will not improve (dis)similarity estimation but only dilute results.

For comparison, we have also computed UMAP embeddings using distances computed from Tanimoto coefficients, see Supplementary Fig. 10. Tanimoto-based plots may also be helpful for spotting inhomogeneous training data; yet, given the known limitations and issues of the fingerprint-based distance measures discussed above, we suggest to treat the resulting UMAP plot with even more care than the MCES-based plot.

## Discussion

Machine learning datasets containing experimental data for molecular structures usually differ substantially from a uniform subset of biomolecular structures. More worrying is the fact that for most datasets, large regions of the biomolecular structure universe remain completely empty. We stress that this is a very practical issue, beyond the "correlation vs. causality" discussion. Using a well-known example from the literature: Even if we accept that childbirths in a country can be predicted from stork population sizes[74], it turns out that the model fails miserably for African countries. Consequently, the absence of any data for all countries outside of Europe should worry us massively on the domain of applicability of the model. We noted above the recent trend to train large end-to-end models without explicitly considering the domain of applicability[1–7]. We note in passing that datasets relying on quantum chemistry calculations instead of experimental values[75], usually do not share this issue.

Several guidelines for good machine learning practice in chemistry and the life sciences were published, due to the increasing importance of machine learning in these areas. For large models trained on small molecules, we suggest including a distribution analysis of the training data into these recommendations; this may indicate whether the trained models are indeed predicting what they are claimed to predict. Otherwise, performance improvements using more complex machine learning models may be an adaption to the

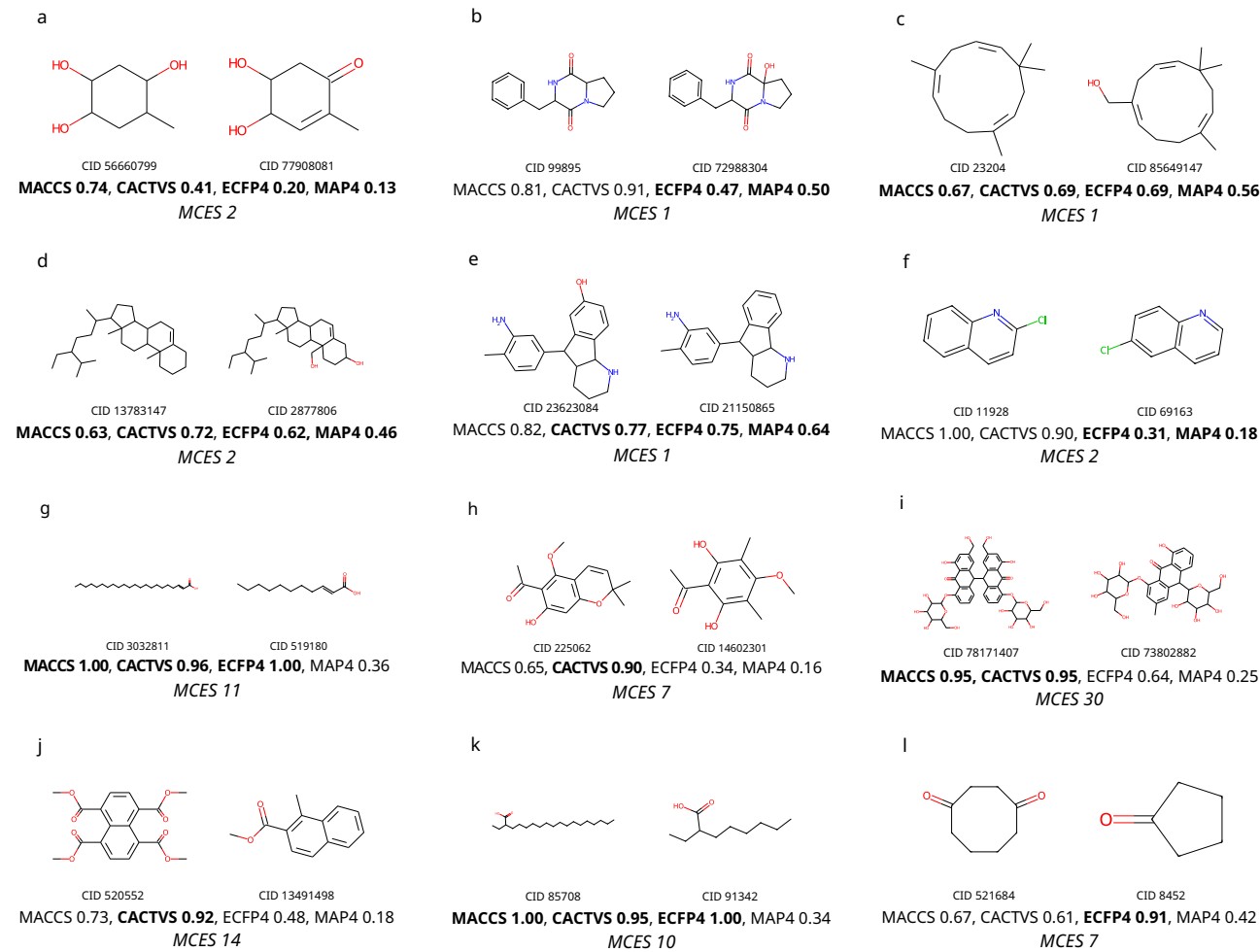

**Fig. 9 | Tanimoto coefficients can differ substantially from perceived structural similarity.** Shown are Tanimoto coefficients for MACCS, CACTVS, ECFP4 (Morgan), and MAP4 fingerprints; high values imply high similarity. The corresponding Jaccard distance is one minus this value. **a–f** Molecular structure pairs with high perceived structural similarity but relatively low Tanimoto coefficient, for one or more fingerprint types. **g–l** Molecular structure pairs with low perceived structural similarity but very high Tanimoto coefficient for at least one fingerprint type. All molecular structures are biomolecular structures; "CID" is the PubChem compound identifier number. It is understood that similar pathological cases exist for another fingerprint type. It is also understood that the problem cannot be solved by using another fingerprint-based similarity measure, such as the Sørensen-Dice coefficient. For comparison, we also report the myopic MCES distance between each molecular structure pair.

peculiarities of the training data, and may potentially result in no improvements in practice.

Our methods may allow us to spot datasets where the distribution of molecular structures is peculiar and potentially hazardous. On the other hand, if a dataset shows no peculiarities, this does not imply a Carte Blanche for machine learning. As an example, consider the MS/MS dataset that, in all of our analyses, did not appear peculiar. Yet, from our own experience with these data, we want to warn the reader that even here, the distribution of molecular structures may result in unexpected behavior of trained models and counterintuitive evaluation results[76,77].

In our analysis, we made a somewhat subjective decisions on what small molecules are "of biological interest". In particular, we did not include the ChEMBL structure database[78], as the distribution of molecular structures differs substantially from our set of biomolecules (Supplementary Fig. 11). We have repeated our analysis of coverage for small training dataset, this time using ChEMBL as our structures of interest (Supplementary Fig. 12). Again, these plots may allow us spot coverage bias and restrictions to the domain of applicability.

We have shown that the presented method for estimating MCES distances exhibits fairly small running times. It is however important to note that the current implementation is Python-based, with hardly any optimizations applied. We conjecture that a C++ implementation of the MCES bounds can reach running times comparable to that of RDKit's RASCAL implementation, as it consists primarily of computing a maximum matching, one of the best-studied problems in theoretical computer science. It is understood that being able to swiftly estimate MCES distances allows for application beyond what we have discussed in this paper. In particular, MCES distances can be used as part of machine learning; notable examples are clustering, $k$-nearest neighbor, and the radial basis function kernel. MCES can also be used to measure the diversity of a set of molecular structures[79,80]. Instead of simply using the Kullback-Leibler divergences[81] for this purpose, we suggest to measure, for every biomolecular structure, its distance to the ($k$-th) closest structure in the training data. As used here, the MCES distance measures an absolute distance between molecular structures; it is understood that it can be readily modified to take into account the sizes of the molecular structures[58]. Similarly, it may be modified to consider substructure relationships. Finally, we may incorporate differences in elements between the two structures where appropriate, for example for halogens.

## Methods

In this paper, we consider only *molecular structures*: Precisely speaking, we consider the identity and connectivity (with bond types including aromatic bonds) of the atoms but ignore the stereo-configuration for asymmetric centers and double bonds. For the sake of brevity, we will simply speak of "molecular structures" in the remainder of this paper. All molecular structures were standardized using the PubChem standardization web service[66]. Few structures that failed standardization were excluded. Tanimoto coefficients in Fig. 9 and Supplementary Fig. 10 were calculated using Molecular ACCess System (MACCS)[68], PubChem (CACTVS)[69], and Extended-Connectivity (ECFP)[70] fingerprints. MACCS and ECFP fingerprints were computed in RDKit, using the "MACCSkeys.GenMACCSKeys" and "rdMolDe-scriptors.GetMorganFingerprintAsBitVect" (radius 2, 2048 bits) functions, respectively; CACTVS fingerprints were retrieved from PubChem. MAP4 fingerprints (MinHashed atom-pair fingerprint up to a diameter of four bonds)[71] were computed with the official Python package (https://github.com/reymond-group/map4), setting the number of bits to 4096 and using 1 minus the value from the "Min-hash.get_distance"-function to obtain Tanimoto coefficients.

### Biomolecular structures

As a proxy of the universe of biomolecules, we use a union of several molecular structure databases that contain such molecules[66,82–95]. Recall that we interpret the term "biomolecules" as molecules of biological interest; this includes primary and secondary metabolites, xenometabolites, drugs, drug degradation products, toxins, but also small molecules from, say, skin care products as well as common contaminants. As *biomolecular structures*, we combine all molecular structures from databases KEGG (Kyoto Encyclopedia of Genes and Genomes), ChEBI (Chemical Entities of Biological Interest), HMDB (Human Metabolome Database), YMDB (Yeast Metabolome Database), PlantCyc, MetaCyc, KNApSAcK, UNPD (Universal Natural Products Database), MaConDa (Mass spectrometry Contaminants Database), HSDB (Hazardous Substances Data Bank), Super Natural II, COCONUT (COlleCtion of Open Natural prodUcTs), and NORMAN (Network of reference laboratories, research centers and related organizations for monitoring of emerging environmental substances). These are databases frequently used in (untargeted) metabolomics, environmental research, and natural products research. See Supplementary Table 1 for details. HMDB and YMDB cover molecules that can be found in specific organisms. KEGG, ChEBI, PlantCyc, MetaCyc, and KNApSAcK allow to link molecules to species and/or pathway information. Common contaminants occurring in MS experiments can be found in MaConDa, hazardous compounds in HSDB. The NORMAN suspect list focuses on compounds that are expected to occur in the environment, including industrial chemicals, pesticides, pharmaceuticals, and food additives. General collections of natural products are UNPD, Super Natural II and COCONUT. COCONUT is a large collection of natural products from 53 different data sources, including ChEBI natural products, KNApSAcK, UNPD, and Super Natural II, but also FooDB, Marine Natural Products, GNPS (Global Natural Products Social) molecular structures, and a subset of ZINC (ZINC Is Not Commercial) with natural products.

In addition, we use subsets of large molecular structure databases that are flagged as being of biological relevance: These are compounds from PubChem which are either MeSH-annotated or part of particular classification schemes based on Schymanski et al.[96]. The PubChem classification "bio and metabolites" comprises structures from the PubChem Compound TOC categories "Biomolecular Interactions and Pathways", "Bionecessity", "Metabolite Pathways", "Metabolism and Metabolites", "Plant Concentrations" and "Metabolite References"; PubChem "drug" comprises structures from "Drug and Medication Information" and "Pharmacology"; PubChem "food" are structures of category "Food Additives and Ingredients"; and PubChem "safety and

toxic" comprises structures of "Toxicity", "Chemical Safety", "Safety and Hazards" and "Agrochemical Information".

Data from all databases was retrieved on February 10th, 2023. For UNDP an older downloaded version was used since the website has been taken offline. The combined dataset contains 718,097 unique molecular structures of metabolites and other molecules that can be expected in biological samples, see again Supplementary Table 1. Notably, 16.7% of our biomolecular structures contain halogens. We argue that this combination of databases may serve as a proxy for known *molecular structures of biological interest*. There are clearly larger databases such as PubChem, but those databases also contain molecular structures not of biological interest.

What small molecules are actually "compounds of biological interest", clearly depends on the research question at hand. For example, we decided not to include compounds from the ChEMBL structure database into our set of biomolecular structures[78]. ChEMBL contains curated data on bioactive molecules (drugs and drug candidates) from the medicinal chemistry literature, as well as data directly deposited to the database. Notably, ChEMBL is larger than our set of "biomolecules", and presently contains 2.4 million compounds. We found that the distribution of molecular structures in ChEMBL differs substantially from the structure databases mentioned above, see Supplementary Fig. 11. Consequently, we decided to keep ChEMBL separated, and use it as an alternative source of structures of interest.

### Molecular structure datasets

We consider the following datasets frequently used in machine learning:

- The *BACE* dataset comprises 1513 synthetic human BACE-1 inhibitors reported in the scientific literature[41]. It provides inhibitory concentrations ($IC_{50}$) and additionally transforms these into binary labels by applying a concentration threshold.
- The *BBBP* dataset consists of 2039 molecules curated from the scientific literature discussing blood-brain barrier penetration[42]. The authors acknowledge that such a dataset curated from the literature is generally biased over-representing positive examples. Modeling blood-brain barrier penetration is of interest since the barrier is impenetrable to most drugs.
- The *ClinTox* dataset compares FDA-approved drugs from SWEET-LEAD database[43] with drugs that failed clinical trials for toxicity reasons retrieved from the Aggregate Analysis of ClinicalTrials.gov (AACT) database (http://www.ctti-clinicaltrials.org/aact-database). The dataset contains 1478 molecular structures.
- The *Delaney* dataset contains 1128 compounds with water solubility data[44].
- The *Lipo* dataset provides lipophilicity data for 4200 compounds[45] and is part of the ChEMBL database[78] which contains bioactive molecules. For each compound octanol-water partition coefficients were experimentally measured (log D at pH 7.4).
- The Side Effect Resource (SIDER)[46] is a database of drugs and adverse drug reactions. The *SIDER* dataset contains 1427 molecular structures assembled in the DeepChem library[47] with side effects grouped in 27 system organ classes according to MedDRA classifications (Medical Dictionary for Regulatory Activities, http://www.meddra.org/).
- The *SMRT* (Small Molecule Retention Time) dataset consists of molecules and their experimentally acquired reverse-phase chromatography retention times[48]. Pure standard materials of 80,038 small molecules including metabolites, natural products, and drug-like small molecules have been measured. This dataset is much larger than the other datasets. To avoid proliferating running times and cluttered plots, we uniformly subsampled 10,000 standardized molecular structures.
- *ToxCast*[49] is an extended dataset from the Tox21 program that includes toxicology data based on in vitro high-throughput

screening of 8576 compounds.

- The *Tox21* dataset consists of samples from 12 toxicological experiments of in vitro bioassays and a total of 7831 molecular structures[50] used in the 2014 Tox21 Data Challenge[51]. This dataset has a relative large overlap in molecular structures with the ToxCast dataset: 6311 molecular structures are found in both datasets.
- The *MS/MS* dataset contains compounds that have at least one reference tandem mass spectrum in a spectral library. It comprises compounds from GNPS[52], MassBank[53], and NIST 17 (commercial, National Institute for Standards and Technology, Tandem Mass Spectral Library, 2017 release). Note that NIST17 is not public but very frequently used to train machine learning models. The dataset underwent several rounds of manual validation, discarding numerous molecular structures in the process[97,98]. In total, the dataset contains 18,848 unique molecular structures. It is different from the other datasets in that it is not one public dataset but rather, a union of different spectral libraries. Furthermore, for each compound one or more tandem mass spectra (MS/MS) are recorded, each being not a single value but rather complex structured data.

All datasets except *SMRT* and *MS/MS* were retrieved from the molecular property prediction framework CHEMPROP[54] at https://github.com/chemprop/chemprop. The *SMRT* dataset was downloaded from https://doi.org/10.6084/m9.figshare.8038913. Structures for *MS/MS* are based on the CSI:FingerID training dataset[97,98]. It was downloaded from https://gnps.ucsd.edu/(GNPS) and https://massbank.eu/(Mass-Bank), but also contains the commercial NIST 17 MS/MS library. See Supplementary Table 2 for further statistics on the datasets. To the best of our knowledge, these are among the largest public datasets containing molecular structures based on *experimental data*. Many of these datasets are living datasets, meaning that more compounds have been added at a later stage. Apparently, Tox21 and ToxCast are now provided as one dataset (https://www.epa.gov/chemical-research/exploring-toxcast-data).

### Subsampling molecular structures

The biomolecular structure set contains 718,097 molecular structures. From this set, we uniformly sampled a subset of cardinality 20,000: Each of the 718,097 molecular structures has exactly the same probability to be drawn; similarly, each subset of cardinality 20,000 has exactly the same probability to be drawn. We later noticed that six of these 20,000 molecular structures are single ions, for example, a single iron ion. These molecular structures resulted in execution errors when computing the MCES, but are also of no interest for our analysis. To this end, we discarded these six structures before computing distances, embeddings, and running times. Consequently, our subsampled set contains 19,994 uniformly sampled molecular structures. The number of nodes $n$ in the subsampled molecular graphs ranges from 1 to 139, with an average of 33.8 nodes (quartiles 20, 28, 40). Recall that we do not consider hydrogen atoms in our computations. The number of bonds (edges without multiplicities) $m$ ranges from 0 to 153, with an average of 35.8 bonds (quartiles 21, 30, 43). The complexity of an instance can be roughly measured by the product $m_1 \cdot m_2$, for number of bonds $m_1, m_2$. This product ranges from 0 to 22,950, with an average of 1279.5 (quartiles 520, 900, 1593). We report quartiles (25%, median, 75%) as our subset contains both very small and very large molecular structures. In the extreme case, a few "molecular structures" consist of a single atom only.

An *instance* consists of a pair of biomolecular structures. To avoid proliferating running times in our running time evaluation, we uniformly subsampled a set of 20,000 instances. To allow that resulting running times are comparable with other computations in this paper, we used the 19,994 molecular structure from above to generate the

instances. We generated 199,870,021 pairs of molecular structures; from this set, we uniformly sampled 20,000 instances. For the sub-sample, the product $m_1 \cdot m_2$ ranges from 0 to 14,124, with an average of 1273.0 (quartiles 525, 910, 1584). The list of molecular structures (both the complete set of biomolecules, the subsample with 19,994 molecular structures, and the 20,000 subsampled structures pairs) is available for download.

### The Maximum Common Edge Subgraph problem

Given two graphs $G_1 = (V_1, E_1)$ and $G_2 = (V_2, E_2)$, their *Maximum Common Edge Subgraph* (MCES) is a graph $G_c = (V_c, E_c)$ with $V_c \subseteq V_i, E_c \subseteq E_i, i \in \{1, 2\}$, such that $|E_c|$ is maximal. We define the *MCES distance* of the two graphs as $|E_1| + |E_2| - 2|E_c|$. Note that a Maximum Common Edge Subgraph minimizes this distance.

Different from other variants of the common subgraph problem, MCES does explicitly not require that the subgraph is connected; otherwise, displacing a single bond may necessitate to exclude almost half of the graphs from the common subgraph. It is easy to see that MCES is NP-hard, as it generalizes subgraph isomorphism which, in turn, can be easily be reduced from the clique problem[99]. See Raymond et al.[26] from 2002 for an early and Englert et al.[58] for a more recent discussion of the different problem variants as well as exact and heuristic algorithms for its solution. Exact methods guarantee that the MCES of two molecular structures is found, despite the computational hardness of the problem. One approach is based on finding a largest clique in the product graph of the line graphs of the two molecular structure input graphs but has to deal with so-called $\Delta Y$ exchanges that happen because of identical line graphs of the two small graphs $K_3$ and $K_{1,3}$. Restriction to finding *connected* common substructures leads to a speed-up, as observed by Koch[100]. It is noteworthy that clique-based algorithms are fastest when the subgraphs and hence, the cliques, are small; hard instances can easily be constructed by considering highly similar molecular structures, which are arguably the most interesting for our application. The program RASCAL[60] is arguably the commercial default for computing the MCES of small molecules based on cliques in a product graph; recently, an implementation of RASCAL has been made available in RDKit (version 2023.09.1). A lesser-known approach, which has so far not been applied to the comparison of molecular structures, is based on integer linear programming[101].

Some algorithms reach improved running times by restricting the input to certain graph classes such as outerplanar graphs[102,103] but are therefore only of limited practical use. Next, certain algorithms do not consider the optimal MCES solution but rather, introduce additional constraints that have to be satisfied, to improve running times. Examples are limiting the number of connected components[28] and considering simplified graph representations[104]. We argue that all of these modifications, which are introduced because of the inability to efficiently compute the MCES rather than problem-immanent considerations, are again of only limited practical use.

We stress that beyond the simple number of edge modifications introduced above, there exist numerous other possibilities to transform the MCES into a distance between molecular structures[60]. We will not discuss this further here, as for the application of mapping all molecular structures into the plane, using absolute distances appears to be the appropriate choice.

### Computing the maximum weight common edge subgraph

We suggest a new approach to compare two molecular structures, which combines the computation of lower bounds as in ref. 60, with an integer linear programming approach based on the formulation given in ref. 101.

In molecular graphs, nodes are labeled by atom type and edges are weighted reflecting their bond order. We therefore compute the *maximum weight common edge subgraph* (MCES), a weighted variant of MCES. Formally, the input consists of two molecular graphs

$M_1 = (G_1, a_1, b_1)$ and $M_2 = (G_2, a_2, b_2)$, where $a_i: V_i \to \Sigma$ denotes the atom type and $b_i: E_i \to \{1, 1.5, 2\}$ specifies the bond type (single, aromatic, or double), for $i \in \{1, 2\}$. The task is to compute the maximum weight common edge subgraph $G_c = (V_c, E_c)$, that is, $V_c \subseteq V_i$ with $a_1(v) = a_2(v)$ for all $v \in V_c$, $E_c \subseteq E_i$ for $i \in \{1, 2\}$, such that their *weighted distance*

$$d(M_1, M_2) = \sum_{e \in E_1} b_1(e) + \sum_{e \in E_2} b_2(e) - 2 \sum_{e \in E_c} \min\{b_1(e), b_2(e)\}$$

is minimal. Here, $\Sigma$ is the set of elements, such as $\Sigma = \{C, H, N, O, P, S, B, F, Si, Cl, Se, Br, I\}$. We reformulate this term and alternatively minimize

$$d(M_1, M_2) = \sum_{e \in E_c} |b_1(e) - b_2(e)| + \sum_{e \in E_1 \setminus E_c} b_1(e) + \sum_{e \in E_2 \setminus E_c} b_2(e).$$

Precisely speaking, we solve the following problem: Given two molecular graphs $M_1$ and $M_2$, and a distance threshold $T$, compute their minimum weighted distance $w(M_1, M_2)$ if this is below $T$ and otherwise an (ideally large) lower bound for $d(M_1, M_2)$.

We solve this problem using an algorithm engineering approach. It consists of first computing lower bounds on the distance. Should the maximum of these bounds already be larger than $T$, we report this maximum. Only if both bounds fail, we use an exact algorithm based on an integer linear program (ILP) to compute $\min\{d(M_1, M_2), T\}$.

The bounds are similar to those used in ref. 60, but we additionally take the weights of the edges into account. The first bound can be computed in linear time using radix sort. It maps nodes onto each other based on their weighted degree. We first sort both node sets $V_1$ and $V_2$ by weighted degree. Then, we iteratively go through both lists by taking and removing the first pair and adding the absolute difference of the weighted degrees to the bound. Should one graph contain more nodes than the other, we also add the weighted degrees of the additional nodes to the bound. In the end, we divide the bound by two, because each edge has been considered twice in this calculation.

The second bound is also based on mapping nodes onto each other, but now we take the atom types of both nodes in an edge into account. We therefore, we construct bipartite graphs with nodes from each atom type on each side. Should the molecules have different numbers of an atom type, we fill up the graphs such that both sides have an equal number of nodes. We now link pairs of nodes in each bipartite graph and set the weight of this edge to the minimum weight to match the two nodes onto each other, which we compute using a complete enumeration approach for mapping the local neighborhoods. We calculate the final bound by computing minimum weight perfect matchings in each of the bipartite graphs, summing up the results and, again, dividing by two. See ref. 60 for correctness proofs. We can compute this bound in cubic time in the number of nodes (heavy atoms), using the Hungarian algorithm[105] or faster methods for finding minimum-weight perfect matchings. In contrast, executing the Integer Linear Program below may require exponential time.

We note that the second bound is stronger than the first, meaning that the value of the first bound can never be larger than that of the second. This follows because a matching for the second bound is also a valid matching for the first bound, but with smaller weight. In addition, we use the same denominator of two for both bounds. The advantage of the first bound is that it can be computed even faster. Since finding minimum matchings can be executed very fast in practice, we concentrated on the second bound throughout this paper.

To compute $\min\{d(M_1, M_2), T\}$, we use an integer linear programming (ILP) formulation. It is similar to the formulation in ref. 101, but explicitly addresses weighted minimization and unmapped edges, and is faster to solve because of the atom labels and the threshold constraint. We introduce variables $y_{ik}$ for each node pair $i \in V_1$ and $j \in V_2$ with $a(i) = a(j)$ to indicate whether $i$ will be mapped to $j$ in which case $y_{ik} = 1$. We introduce similar variables $c_{ijkl}$ for mappable edge pairs

$ij \in E_1$ and $kl \in E_2$ with $a(i) = a(j)$ and $a(k) = a(l)$ with corresponding weights $w_{ijkl} = |b(ij) - b(kl)|$. Finally, we have variables $n_{ij}$ for all $ij \in E_1 \cup E_2$ to indicate whether an edge is not mapped. We denote by $N(i)$ the neighborhood of node $i$, that is the set of adjacent nodes.

Our ILP is as follows:

$$\min \quad \sum_{ijkl} c_{ijkl} w_{ijkl} + \sum_{ij \in E_1 \cup E_2} n_{ij} b(ij) \tag{1}$$

$$\text{s. t.} \quad \sum_{k \in V_2} y_{ik} \leq 1 \qquad \text{for all } i \in V_1 \tag{2}$$

$$\sum_{i \in V_1} y_{ik} \leq 1 \qquad \text{for all } k \in V_2 \tag{3}$$

$$\sum_{j \in N(i)} c_{ijkl} \leq y_{ik} + y_{il} \qquad \text{for all } i \in V_1, \text{ for all } kl \in E_2 \tag{4}$$

$$\sum_{l \in N(k)} c_{ijkl} \leq y_{ik} + y_{jk} \qquad \text{for all } k \in V_2, \text{ for all } ij \in E_1 \tag{5}$$

$$\sum_{kl \in E_2} c_{ijkl} + n_{ij} = 1 \qquad \text{for all } ij \in E_1 \tag{6}$$

$$\sum_{ij \in E_1} c_{ijkl} + n_{kl} = 1 \qquad \text{for all } kl \in E_2 \tag{7}$$

$$\sum_{ijkl} c_{ijkl} w_{ijkl} + \sum_{ij \in E_1 \cup E_2} n_{ij} b(ij) \leq T \tag{8}$$

$$y_{ik} \in \{0, 1\} \qquad \text{for all mappable } i \in V_1 \text{ and } k \in V_2 \tag{9}$$

$$c_{ijkl} \in \{0, 1\} \qquad \text{for all mappable } ij \in E_1 \text{ and } kl \in E_2 \tag{10}$$

$$n_{ij} \in \{0, 1\} \qquad \text{for all } ij \in E_1 \cup E_2 \tag{11}$$

We implemented the approach using Python with packages `networkx` for graphs and graph algorithms and `pulp` for the ILP solver interface, which enables the usage of different optimizers like CPLEX, Gurobi or free libraries. For further details of our approach and implementation, see ref. 62.

We stress that we can stop ILP computations at basically any time, and receive both upper and lower bounds for the objective function from the ILP solver. Similarly, we can start the ILP solver with a running time limit such as 5 min. This prohibits that few instances require hours of computation time, but nevertheless, provides a lower of bound for the myopic MCES distance.

## Running time evaluations

All running times were measured on a 40-core Intel Xeon E5-2698 2.20GHz processor running 80 threads in parallel. Python version 3.9.7 on Ubuntu 20.04 was used to run the scripts; ILP computations were executed with CPLEX version 12.8 (https://www.ibm.com/support/pages/cplex-optimization-studio-v128). Running times of individual instances were estimated via wall clock time using Python's "time" module. We stress that the resulting running times are not optimized, in the sense that we did not exclude interrupts, thread switching, etc. Furthermore, running two threads on a single core via hyperthreading reduces the overall running time of the complete batch, but increases the running time of each thread. Yet, the running times reported here are a good measure of what to expect when computations are

executed in applications, where such running time optimizations are uncommon.

Notably, different settings were applied in the running time evaluations. For the complete set of instances, CPLEX was executed via `pulp` with the default settings. In this mode, CPLEX may use all cores available, potentially starting a large number of threads. Especially when already computing instances in parallel, this leads to suboptimal running times − both because of the overhead introduced by starting a large number of threads and because threads started by CPLEX may block other computations. For the subsampled set of 20k instances, CPLEX was executed with the parameter "threads" set to 1, restricting CPLEX to single-threaded mode. This option is only available when using the "CMD"-version of CPLEX in `pulp`.

For comparisons with RASCAL, the RDKit implementation in version 2023.09.3 was used; the time limit was set to one hour. As described above, for part of the comparison the similarity thresholds were modified. To obtain approximations when no exact computations were performed, the option "returnEmptyMCES" was enabled. To circumvent the internal limit on the size of the product graph, the option "maxBondMatchPairs" in the C++ base of RDKit was exposed to Python interface, necessitating recompilation. Otherwise, all settings were left at their default values.

### The myopic MCES distance is a metric

We now prove that the myopic MCES distance is a metric. The proof proceeds in four steps: For completeness, we show that the first bound gives rise to a pseudometric. Next, we show that the second bound is a pseudometric, too. Third, we show that the exact MCES distance is a metric. Finally, we use all of these results to show that the myopic MCES distance is a metric, for all $T > 0$.

The first bound equals zero if the node set and weighted degrees are identical. It is symmetric because the absolute difference is symmetric. To see that the triangle inequality holds, we note that this bound is in fact the weight of a minimum matching between the two node sets. Now, two matchings between $V_1$ and $V_2$, and $V_2$ and $V_3$, respectively, can be combined into a matching between $V_1$ and $V_3$. Every edge weight is at most as large as the sum of edge weights in the two original matchings. Hence, the minimum matching between $V_1$ and $V_3$ will have weight at most as large as the sum of matching weights. Next, the same arguments also hold for the second bound, which, again, corresponds to a minimum weight matching, albeit with more sophisticated edge weights. Let $d_B$ be the second bound.

Third, for the exact MCES distance $d_E$, it is understood that the weighted distance between two molecular graphs is symmetric and that it is zero if and only the molecular graphs are identical. Proving the triangle inequality for the weighted MCES distance can be done analogously to the proof of Bunke et al.[106] who showed that a related MCS distance is a metric.

Fourth, we have shown that the bound $d_B$ is a pseudometric and that the exact distance $d_E$ is a metric. We want to show that the myopic MCES distance $d$ is a metric, too. It is straightforward to infer from the above that $d$ is symmetric. Similarly, $d(x, x) = 0$ is clear. Next, $d(x, y) = 0$ implies $d_B(x, y) = 0 < T$ and, hence, $0 = d(x, y) = d_E(x, y)$. Since $d_E$ is a metric, we infer $x = y$. Hence, what remains to be shown is that $d$ fulfills the triangle inequality $d(x, z) \leq d(x, y) + d(y, z)$ for all $x, y, z$.

Recall that $d(x, y) = \min\{d_E(x, y), T\}$ in case $d_B(x, y) < T$. Clearly, $d(x, y) = d_B(x, y) \geq T$ if $d_B(x, y) \geq T$. We stress that $d_B(x, y) \leq d(x, y)$ must hold even if $d(x, y) = \min\{d_E(x, y), T\}$: In this case, $d_B(x, y) < T$ and $d_B(x, y) \leq d_E(x, y)$ holds.

We make a case distinction on whether $d(x, z)$ was computed via the ILP or via bounds. (i) Assume $d(x, z) = \min\{d_E(x, z), T\}$ holds. If at least one of the two distances $d(x, y)$ and $d(y, z)$ is at least $T$

(say, $d(x, y) \geq T$) then

$$d(x, z) = \min\{d_E(x, z), T\} \leq T \leq d(x, y) \leq d(x, y) + d(y, z). \quad (12)$$

This covers the case that one of the distances $d(x, y)$ and $d(y, z)$ was computed heuristically. So, $d(x, y) = d_E(x, y)$ and $d(y, z) = d_E(y, z)$. Now,

$$d(x, z) = \min\{d_E(x, z), T\} \leq d_E(x, y) \leq d_E(x, y) + d_E(y, z) = d(x, y) + d(y, z)$$

since $d_E$ is a metric. (ii) Now, assume $d(x, z) = d_B(x, z)$ was computed heuristically. But then, $d(x, z) = d_B(x, z) \leq d_B(x, y) + d_B(y, z) \leq d(x, y) + d(y, z)$. □

The important step in this proof is (12): Here, we require that $d$ is double thresholded via $d(x, z) = \min\{d_E(x, z), T\}$. If we do not use double thresholding, then the resulting distance measure is not a metric, see Supplementary Fig. 8.

### Reporting summary

Further information on research design is available in the Nature Portfolio Reporting Summary linked to this article.

## Data availability

Data used throughout this paper including the biomolecular structures dataset, the *MS/MS* dataset as well as computed MCES distances and corresponding UMAP embeddings are available at https://github.com/boecker-lab/myopic-mces-data. Interactive UMAP visualizations are available at https://mces-data.boeckerlab.uni-jena.de/. The biomolecular structures dataset combines structures from the databases KEGG[82], ChEBI[83], HMDB[84], YMDB[85], PlantCyc[86], MetaCyc[87], KNApSAcK[88], UNPD[89], MaConDa[90], HSDB[91], Super Natural II[92], COCONUT[93], NORMAN[94], and subsets from PubChem[66], retrieved on February 10th, 2023. For UNDP an older downloaded version was used since the website has been taken offline. ChEMBL[78] was accessed on July 31st, 2024. Molecular structures of the training datasets except *SMRT* and *MS/MS* are available from https://github.com/chemprop/chemprop. The *SMRT* dataset is available at https://doi.org/10.6084/m9.figshare.8038913. Source data are provided with this paper and via figshare at https://doi.org/10.6084/m9.figshare.27203583.

## Code availability

Code is open source and freely available at https://github.com/AlBi-HHU/myopic-mces and from Zenodo (https://zenodo.org/records/13912999)[107].

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

## Acknowledgements

F.K. and S.B. supported by Deutsche Forschungsgemeinschaft (BO 1910/23) and by the Ministry for Economics, Sciences and Digital Society of Thuringia (framework ProDigital, DigLeben-5575/10-9).

## Author contributions

S.B. designed the research. F.K. executed all computations and evaluations and prepared all figures. M.L. assembled the set of biomolecular structures and supported evaluations. J.S., G.K., and S.B. developed the method for computing myopic MCES distances, and J.S. implemented the method. S.B. and G.K. wrote the manuscript, with the help of the other authors.

## Funding

## Competing interests

The authors declare no competing interests.
