## [Peer Review file · Nature Communications]

Coverage bias in small molecule machine learning

Corresponding Author: Professor Sebastian Böcker

Version 0:

Reviewer comments:

Reviewer #1

(Remarks to the Author)

The article starts with a provocative title, which sadly the article fails to support. This is really a paper about an implementation of the MCES approach: if the authors focussed just on this aspect, and in showing that it forms an effective similarity metric, it would be worth publishing, but in a specialist journal. The authors should take the work of Rarey they cited (plus Maximum Common Substructure Searching in Combinatorial Make-on-Demand Compound Spaces Robert Schmidt, Raphael Klein, and Matthias Rarey

Journal of Chemical Information and Modeling 2022 62 (9), 2133-2150

DOI: 10.1021/acs.jcim.1c00640) as template of how to do this. It is obvious that the authors have not much experience in QSAR as practiced in Pharma and Agro, and are targeting only academic papers. There is a huge difference between 'just about publishable' and what is used to decide work in the real world.

The fallacy that the authors fall into, as exemplified by the use of the stork model analogy, is that activity is distributed across all of chemical space. This is true for solubility, but not for biological activity. A serotonin transporter has evolved to recognise serotonin, so one would not expect any observable data for, say, the steroid class. Does that mean that the model is not useful? No. It means that the model has an applicability domain. If biological function did not value selectivity, life would not exist. It is not intended to cover all small molecules of biological interest. The authors use arguments bordering on vitalism to define this term. What about all the data in ChemBI? What about all the data published in J. Med. Chem. showing synthetic compounds having activity way better than the endogenous ligand? Yet these compounds are excluded.

The use of substructure based fingerprints is deprecated by most cheminformaticians - Morgan fingerprints are preferred. MACCS keys are not rich enough. The examples in Fig S 11 suffer from this. Fingerprints only make sense when applying the similarity principle, that similar compounds have similar profiles. Nothing is said about dissimilar molecules. Below a certain level, the metric cannot be used to say that one molecule is more similar to the query than another, as the authors themselves acknowledge.

There are some standard best practices about train/test/validate that the authors seem to be unaware of (Time-Split Cross-Validation as a Method for Estimating the Goodness of Prospective Prediction.

Robert P. Sheridan

Journal of Chemical Information and Modeling 2013 53 (4), 783-790

DOI: 10.1021/ci400084k). And other works by this author, including work on when it not worth adding any more data to the model.

In the Ertl paper, there is large overlap between synthetic molecules and natural products, so it is not a useful way to discriminate molecules of no biological interest. See also the comments about solubility, and add in MS/MS, logP as assays that are physical, not biological. Which is why the MS/MS dataset did not appear peculiar.

For the MCES method, comparison is needed with existing algorithms.

In conclusion, this paper does not give any real insight into the issues of data set construction or model building. The graph-based distance from MCES algorithm might be useful, but it is crippled by the run-times.

Reviewer #2

(Remarks to the Author)

Note: To make this review easier to process, I have chosen to format it using markdown.

In this manuscript, the authors show that data sets which are commonly used in machine learning for chemistry and biochemistry do not cover a broad chemical space and may therefore not be applicable beyond these largely synthetic benchmarks. To analyse this coverage, the authors introduce a new method to calculate the maximum common edge subgraphs of molecular graphs, which allows them to use this similarity metric to compare the data sets in question with a larger data set more representative of the complete chemical space (that is of biological interest).

I believe the paper makes an important contribution that is of interest to the machine learning community in biochemistry and chemistry. The problem regarding the used data sets, or benchmarks, is a timely topic and is currently not as discussed as widely as it should (for a good current discussion, see e.g. <http://practicalcheminformatics.blogspot.com/2023/08/we-need-better-benchmarks-for-machine.html>)

Abstract

- "Applications include prediction of toxicity, ligand binding or retention time."
- Rather than mentioning a specific example such as retention time, it may be more informative to mention "pharmacokinetic properties" as a whole
- "We investigate how well certain large-scale datasets cover the space of all known biomolecular structures."
- As biomolecular structures include macromolecules, do you agree it would be better to replace it with "molecular structures found in the metabolome", which would exclude macromolecules?
- "this computational problem is NP-hard"
- Given the general nature of the journal, it may be better to not use computer science jargon in the abstract and describe it as non-trivial rather than NP-hard.
- As an aside, isn't MCES NP-complete? Not that NP-hard is wrong, but it would be more specific

Introduction

- "Whereas it is comparatively simple to train a machine learning model that performs well in evaluations, it is much harder to derive a model that indeed contributes to solving the underlying question such as toxicity prediction."
- Here, would it not be better to directly introduce the problem of generalisation and then give toxicity prediction as an example?
- "We argue that with the advent of deep learning, the trenches between models claimed to perform well and those that perform well in practice have broadened."
- It would be beneficial to shortly discuss issue of interpolation vs extrapolation for generative models, especially in the context of an incomplete coverage of the true distribution in the given training data. A recent manuscript that discusses this issue (of, ultimately, generalisation) for transformer-based models that have become heavily used in chem- and bioinformatics is <https://arxiv.org/abs/2311.00871>.
- Last paragraph of the introduction
- An additional aspect of chemical spaces that could be discussed is the concept of activity cliffs (<https://www.sciencedirect.com/science/article/abs/pii/S1359644614000361>) and whether MCS-based approaches may provide an advantage over fingerprint-based methods in that regard.
- "require to solve computationally hard problems"
- Given the general audience of the journal, it may be appropriate to briefly describe what makes this problem computationally hard.

Results

- "biomolecular structures"
- As with my comment regarding the abstract, I wonder whether the term "biomolecular" is too broad. Maybe "small-molecule structures of biological interest" or something else along these lines may be better to make again clear that this excludes macromolecules.

Distribution of biomolecular structures

- "As used here, the union of databases contains 718,097 biomolecular structures,"
- Currently, the ChEMBL database contains ~2.4 million compounds of biological interest. Was there a reason this source was not included?
- "To speed up computations, we estimated (provably correct) lower bounds of all distances. We performed exact computations only if the distance bound is at most a chosen distance threshold, which we set to 10, unless stated otherwise."
- I have two comments in regard to this: (i) The proof should be referenced in the methods, and (ii) what is the theoretical and practical speed-up?
- "t-SNE and Minimum Spanning Trees visualizations are provided as examples in Supplementary Fig. 13."
- A reference (or a description of the implementation) for the MST method is missing.
- "To avoid both proliferating running times and cluttered plots, we subsampled 20,000 biomolecular structures"
- The method for subsampling should be described in detail.
- Fig. 1. Map of biomolecular structures with color-coded compound classes.
- For comparison, an embedding of a fingerprint space should be included, as the introduction highlights this as an alternative approach. As metabolites are the focus, using the MAP4 fingerprint may be prudent (<https://pubmed.ncbi.nlm.nih.gov/33431010/>).
- "If we are able to spot non-uniformness in the 2-dimensional UMAP embedding, then it is presumably not a uniform subsample in higher dimensions, either."
- Why evaluate the uniformness of the subsample on the UMAP embedding?

Distribution of molecular structures in public datasets

- "We consider ten public molecular structure datasets frequently used to train machine learning models"
- It may be important to note, that increasingly often, these sets are "only" used to evaluate models trained on larger data sets (see all the recent transformer-based models) with and without fine-tuning.

- "We argue that most of the public datasets are also not representative, meaning that large areas of the biomolecular structures are completely missing in the datasets. In fact, some datasets are concentrated in one or few areas in the plot."
- As noted in the previous point, many models, or more specifically architectures (which may be another good point to make, that certain architectures may only work well in specific areas of the chemical space, AFAIK, this has not yet been explored), are merely evaluated on these data sets. It may therefore be interesting to also show the combined coverage of these "moleculenet" sets in an additional plot.

Tanimoto coefficients

- "molecular structure into a bit string of fixed length"
- A binary array (or vector) may be a better description than a "bit string".
- Furthermore, it should be explained here which method (ECFP?) was used, as the results may differ depending on the chosen method.
- "using the Jaccard index or the Jaccard distance for similarities and dissimilarities"
- The connection between Jaccard distance and the Tanimoto coefficient should be discussed.
- "For completeness, we have also computed an UMAP embedding using distances computed from Tanimoto coefficients"
- As noted previously, the chosen fingerprint plays a much bigger role than the chosen distance function (Jaccard, in this case).

Maximum Common Edge Subgraph computations

- "Our method is the first to use an ILP for the comparison of chemical structures and, to the best of our knowledge, is also the first public publicly available implementation for exact MCES computations."
- I am also not aware of an existing implementation with this approach.
- "For the ILP, 24 of 20,000 instances did not finish within four days of wall clock time."
- It would be interesting to show these structures in the SI.
- "To exclude bias through subsampling, we repeated the above analysis using all pairs from the 19,994 biomolecular structures."
- Where does this number (19,994) come from? By reading the manuscript, I have not encountered it so far.

Compound class distribution

- General note
- It should also be mentioned, that recent model evaluations on these data sets use what is known as Murcko scaffold splits. These account, at least partially, for the issues mentioned, as performance impacts of scaffolds that are not part of the training set are accounted for.

Natural product-likeness score distributions

This is a valuable analysis that may even be extended to other properties in a future publication.

Methods

- "we consider only "two-dimensional" molecular structures"
- "molecular topology" may be a better wording, as two-dimensional structure implies an embedding.

Biomolecular structures

- "There are clearly larger databases such as PubChem, but those databases also contain molecular structures not of biological interest"
- As mentioned previously, ChEMBL would contain only molecules of biological interest.

Molecular structure datasets

No comments.

Subsampling molecular structures

- "From this set, we uniformly sampled a subset of cardinality 20,000".
- As previously noted in the main part, the nature of the uniform sampling is not clear (at least to me). In respect to which property of the molecules was the sampling uniform?
- "We later noticed that six of these 20,000 molecular structures are single ions, for example, a single iron ion."
- I noted previously that it would be good to know where the number "19,994" comes from. As the same will probably happen to other readers, there should be a reference to this methods subsection in the main text or a short explanation should be added to the main.

The Maximum Common Edge Subgraph problem

No comments.

Computing the Maximum Weight Common Edge Subgraph

I also have no specific comments on this section. However, I feel like the whole section, with the exception of the detailed description of the ILP, should be in the Results section. On the other hand, the section is fairly technical. So I leave this for the authors to consider.

Running time evaluations

No comments.

The myopic MCES distance is a metric

No comments.

Conclusion

I agree with this conclusion.

Version 1:

Reviewer comments:

Reviewer #1

(Remarks to the Author)

I am glad to say see that many changes have been made. I would also add it never a good idea to insult the reviewer by assuming that I did not know the Box quote. The main issue I still have is with the definitions of 'biomolecular structures of interest' or ' molecules of no particular biological interest': this is defined by the datasets employed, and not by any formal definition such as showing efficacy in man, or found as a natural product, or showing in vitro activity above a threshold. As such, bias has already been introduced, and this should be acknowledged. The whole Pharma business is based on "synthetic molecules"! It is known that halogens are uncommon in natural products, for example, but may be rescued by the NIST or solubility datasets. There are some pointers here <https://www.peter-ertl.com/publications.html> for example reprint 138.

Reviewer #3

(Remarks to the Author)

The authors describe an analysis of molecular datasets. They visualize the chemical space of these datasets by computing a UMAP of distances constructed from a "myopic MCES" metric. The method is interesting and there is potential here, but ultimately what is shown is that different molecular datasets are drawn from different distributions, which isn't particularly surprising or noteworthy (histogramming molecular weights would reach the same conclusion in most cases).

This could be a much more interesting and useful paper if the authors used their method to illustrate the extent various splitting methods (e.g. scaffold split, fingerprint similarity split) are (or are not) effective at constructing truly independent test sets that are an effective measure of the generalizability of trained models.

I do not understand the authors' resistance to including ChEMBL. If a compound is active in a biological assay it is clearly biologically relevant. The fact that ChEMBL compounds apparently have a different distribution from the other molecules considered only strengthens the claim that the space of biologically relevant molecules is much larger than commonly used training sets.

The authors lament that end-to-end models neither consider or analyze their domain of applicability, yet do not give best practices on how to do that. There seems to be an assumption that only chemical similarity can be used to define a domain of applicability, but this isn't true. A stronger paper might select a state-of-the-art property prediction model that includes confidence scores and map how those scores relate to distance from the training set in UMAP space or demonstrate a relationship between error and distances in MCES distance from the training set.

Overall, the methodology is sound and has potential, but the results are not particularly noteworthy.

Version 2:

Reviewer comments:

Reviewer #1

(Remarks to the Author)

I still have qualms about the dataset definitions, but the authors have at least acknowledged the arbitrary nature of their definitions in the m/s.

Reviewer #3

(Remarks to the Author)

The authors have made minor improvements. My overall impression that that the methodology is sound and has potential, but the results are not particularly noteworthy remains.

Reviewer #1 (Remarks to the Author):

The article starts with a provocative title, which sadly the article fails to support.

The title of the paper is not that provocative, given that it plays with the proverbial “all models are wrong” quote of George Box, see https://en.wikipedia.org/wiki/All_models_are_wrong. We assumed that everybody using machine learning or statistics was familiar with this quote. Apparently, we were wrong. Compare to the much harsher title “All models are wrong and yours are useless: making clinical prediction models impactful for patients” by Florian Markowetz in *npj Precision Oncology* 2024, see here.

Notwithstanding, we have changed the title of our manuscript to “**Coverage bias in small molecule machine learning**”. We would be very happy to reintegrate Box’s quote into the title, but we do not insist.

We show that end-to-end models cannot predict, say, the toxicity of small biomolecules, simply because the training data are insufficient. These end-to-end models **do not define any domain of applicability** and, hence, they are not useful in their current form. To this end, we must assume that the “fails to support” refers to the “all models are wrong” part. Yet, this is merely a quote from one of the smartest statisticians ever.

This is really a paper about an implementation of the MCES approach: if the authors focussed just on this aspect, and in showing that it forms an effective similarity metric, it would be worth publishing, but in a specialist journal.

This would not improve the manuscript but rather make it smaller and, hence, easier to reject.

The authors should take the work of Rarey they cited as template of how to do this.

We have published numerous papers in computational method development, algorithm engineering and bioinformatics. We believe that the evaluation of the MCES method provided in the manuscript, is highly detailed, very informative and also easy to understand. If the reviewer has detailed suggestions on how we can further improve this part of the manuscript, then we are happy to follow them.

It is obvious that the authors have not much experience in QSAR as practiced in Pharma and Agro, and are targeting only academic papers. There is a huge difference between 'just about publishable ' and what is used to decide work in the real world.

We do not know how to address this comment: We indeed want to publish an academic paper here, and consequently, we are indeed targeting “only” academic papers. What is and what is not “practiced in pharma and agro”, seemingly behind closed doors, can indeed not be the subject of an academic paper.

The fallacy that the authors fall into, as exemplified by the use of the stork model analogy, is that activity is distributed across all of chemical space. This is true for solubility, but not for biological activity. A serotonin transporter has evolved to recognise serotonin, so one would not expect any observable data for, say, the steroid class. Does that mean that the model is not useful? No. It means that the model has an applicability domain. If biological function did

not value selectivity, life would not exist. It is not intended to cover all small molecules of biological interest. The authors use arguments bordering on vitalism to define this term.

There is a recent trend in small molecule machine learning to use end-to-end models, **without defining a domain of applicability** and without ever investigating this domain. Numerous high-end publications in *Nature*, *Science* and *Cell* demonstrate this trend. Noteworthy examples are generative models for novel antibiotics and highly toxic small molecules, or classifiers for antibiotic activity, olfactory perception and enzyme-substrate prediction. Furthermore, thousands of machine learning papers use the data from the MoleculeNet paper, without wasting a thought on the distribution of the training or evaluation data. In comparable situations such as image recognition, it has turned out that machine learning models with improved evaluation statistics did not perform any better in practice, but rather started learning the “quirks” of the data distribution.

Apparently, this fact was not clear from the abstract and introduction of our manuscript. Consequently, **we have completely rewritten the introduction and the corresponding part of the abstract.**

What about all the data in ChemBI? What about all the data published in J. Med. Chem. showing synthetic compounds having activity way better than the endogenous ligand? Yet these compounds are excluded.

We do not understand this comment.

The use of substructure based fingerprints is deprecated by most cheminformaticians - Morgan fingerprints are preferred. MACCS keys are not rich enough. The examples in Fig S 11 suffer from this.

Morgan fingerprints **are** substructure-based fingerprints, and are identical to ECFP fingerprints already covered in the manuscript. The reviewer might refer to combinatorial fingerprints vs. those defined via explicit lists of substructures (MACCS, CACTVS). We cover both types of fingerprints in our manuscript. **We now mention that Morgan fingerprints and ECFP fingerprints are basically identical. In addition, Supplementary Fig. 14 now also covers MAP4 fingerprints, see below.**

Fingerprints only make sense when applying the similarity principle, that similar compounds have similar profiles. Nothing is said about dissimilar molecules. Below a certain level, the metric cannot be used to say that one molecule is more similar to the query than another, as the authors themselves acknowledge.

We agree, but we do not see how this interferes with any statements in our manuscript.

There are some standard best practices about train/test/validate that the authors seem to be unaware of (Time-Split Cross-Validation as a Method for Estimating the Goodness of Prospective Prediction). And other works by this author, including work on when it not worth adding any more data to the model.

There are also numerous machine learning papers on this subject, but this has nothing to do with what we discuss in our manuscript. If all available data have the same distribution bias, then no cross-validation or data splitting can ever detect that. **We now mention this fact in the introduction.** Time-split cross-validation cannot help unless complete compound classes

are added to datasets at a certain point in time. Also, the MoleculeNet datasets are no databases with timestamps, so time-splitting is not possible.

In the Ertl paper, there is large overlap between synthetic molecules and natural products, so it is not a useful way to discriminate molecules of no biological interest. See also the comments about solubility, and add in MS/MS, logP as assays that are physical, not biological. Which is why the MS/MS dataset did not appear peculiar.

We do not claim that the NP-likeness score can differentiate, for a single molecule, whether it is of biological interest or not. Yet, we found that the distribution of scores is a good indicator that the distribution of molecular structures differs substantially between two datasets. *We have added a sentence to the manuscript.*

For the MCES method, comparison is needed with existing algorithms.

Until recently, there were no existing algorithms for MCES, except for the commercial RASCAL implementation. There exist certain “modified versions of MCES”, but a comparison would be highly involved and, in our eyes, of limited interest. We mention this in the manuscript. *There is now a free implementation of the RASCAL method as part of RDKit, and we have executed our evaluations for this implementation, too.* We observe the expected problems: RASCAL is based on finding cliques in the product graph. For highly similar & large molecules, this approach results in excessive memory usage, exploding running times, or both. This is to be expected for an NP-hard problem.

In conclusion, this paper does not give any real insight into the issues of data set construction or model building. The graph-based distance from MCES algorithm might be useful, but it is crippled by the run-times.

We emphatically disagree on both points. *With regards to running times, we now mention in the discussion that computing the MCES bounds can be made much faster by switching from the interpreted Python implementation to a compiled C++ implementation.* We refrained from doing so because for an NP-hard problem, such low level optimizations should only be done after one has established that the “hard instances” can be solved in reasonable time. With our manuscript, we have established this fact. We have recently started a small project to speed up the bound computations, by switching to a different programming language.

Reviewer #2 (Remarks to the Author):

In this manuscript, the authors show that data sets which are commonly used in machine learning for chemistry and biochemistry do not cover a broad chemical space and may therefore not be applicable beyond these largely synthetic benchmarks. To analyse this coverage, the authors introduce a new method to calculate the maximum common edge subgraphs of molecular graphs, which allows them to use this similarity metric to compare the data sets in question with a larger data set more representative of the complete chemical space (that is of biological interest). I believe the paper makes an important contribution that is of interest to the machine learning community in biochemistry and chemistry. The problem regarding the used data sets, or benchmarks, is a timely topic and is currently not as discussed as widely as it should (for a good current discussion, see e.g. <http://practicalcheminformatics.blogspot.com/2023/08/we-need-better-benchmarks-for-machine.html>)

Many thanks for the positive feedback. The blog post was an excellent read, many thanks for the link. We fully agree with Pal Walters and have added a citation. Finally, many thanks for the comments below, which considerably improved the manuscript and its readability.

"Applications include prediction of toxicity, ligand binding or retention time": Rather than mentioning a specific example such as retention time, it may be more informative to mention "pharmacokinetic properties" as a whole

We have added "pharmacokinetic properties" in the abstract. We would prefer to leave the example of retention time prediction for small molecules, as a whopping 2880 papers have been written that address exactly this single question. It is somewhat mind boggling.

Articles About 2,880 results (0,19 sec) My profile My library

Any time Since 2024 Since 2023 Since 2020 Custom range...

Sort by relevance Sort by date

Any type Review articles

include patents include citations Create alert

[HTML] Machine learning to predict retention time of small molecules in nano-HPLC [HTML] springer.com Full View

S Osipenko, I Bashkirova, S Sosnin, O Kovaleva... - Analytical and ..., 2020 - Springer

... In our work, we describe a complex approach to predict retention times for nano-HPLC that ... the METLIN small-molecule dataset and simple projection of the results with a small number ...

☆ Save Cite Cited by 28 Related articles All 9 versions Web of Science: 21

[HTML] The METLIN small molecule dataset for machine learning-based retention time prediction [HTML] nature.com Full View

X Domingo-Almenara, C Guijas, E Billings... - Nature ..., 2019 - nature.com

... for retention time prediction lack sufficient accuracy due to limited available experimental data. Here we introduce the METLIN small molecule ... covering up to 80,038 small molecules. To ...

☆ Save Cite Cited by 137 Related articles All 10 versions Web of Science: 99

[PDF] Comprehensive and empirical evaluation of machine learning algorithms for small molecule LC retention time prediction [PDF] acs.org Full View

R Bouwmeester, L Martens, S Degroeve - Analytical chemistry, 2019 - ACS Publications

... all mass spectrometric analyses of (bio)molecules. Because of the high-throughput nature of ... evaluation of machine learning algorithms for retention time prediction is needed to find a ...

☆ Save Cite Cited by 73 Related articles All 6 versions Web of Science: 50

"We investigate how well certain large-scale datasets cover the space of all known biomolecular structures." As biomolecular structures include macromolecules, do you agree it would be better to replace it with "molecular structures found in the metabolome", which would exclude macromolecules?

We have rephrased the abstract, we believe it is now clear that we are talking about small molecules only.

"This computational problem is NP-hard": Given the general nature of the journal, it may be better to not use computer science jargon in the abstract and describe it as non-trivial rather than NP-hard.

Done.

As an aside, isn't MCES NP-complete? Not that NP-hard is wrong, but it would be more specific

We have added "As MCES is clearly in NP, it is NP-complete" to the manuscript. Yet, the interesting part is that the problem is NP-hard. Proofs that some problem is in NP usually read, "it is clear that the problem is in NP"; one of the few exceptions we know of is PRIME, which in the end turned out to be in P. It also does not have any practical consequences that a problem is in NP.

"Whereas it is comparatively simple to train a machine learning model that performs well in evaluations, it is much harder to derive a model that indeed contributes to solving the underlying question such as toxicity prediction." - Here, would it not be better to directly introduce the problem of generalisation and then give toxicity prediction as an example?

Abstract reformulated.

"We argue that with the advent of deep learning, the trenches between models claimed to perform well and those that perform well in practice have broadened." It would be beneficial to shortly discuss issue of interpolation vs extrapolation for generative models, especially in the context of an incomplete coverage of the true distribution in the given training data. A recent manuscript that discusses this issue (of, ultimately, generalisation) for transformer-based models that have become heavily used in chem- and bioinformatics is <https://arxiv.org/abs/2311.00871>.

We have expanded our discussion on generalization accordingly, adding a complete paragraph to the introduction ("The problem of generalization within a dataset..."). For the transformer-based models, we believe that this is very interesting research, but we would argue that it is leading a little too far away from the story we are telling here.

An additional aspect of chemical spaces that could be discussed is the concept of activity cliffs (<https://www.sciencedirect.com/science/article/abs/pii/S1359644614000361>) and whether MCS-based approaches may provide an advantage over fingerprint-based methods in that regard.

Discussion on activity cliffs has been added. We actually do not know whether MC(E)S-based methods will offer an advantage here, so this would be extremely speculative.

"Require to solve computationally hard problems": Given the general audience of the journal, it may be appropriate to briefly describe what makes this problem computationally hard.

We have added a footnote explaining computational hardness in much detail.

"Biomolecular structures": As with my comment regarding the abstract, I wonder whether the term "biomolecular" is too broad. Maybe "small-molecule structures of biological interest" or something else along these lines may be better to make again clear that this excludes macromolecules.

This is in fact how we wanted to define "biomolecular structures", and we have added that description to the definition. Yet, for the sake of readability and brevity, we would prefer sticking with "biomolecular structure" for the rest of the manuscript.

"As used here, the union of databases contains 718,097 biomolecular structures": Currently, the ChEMBL database contains ~2.4 million compounds of biological interest. Was there a reason this source was not included?

We found that the distribution of molecular structures of the 1.9 million small molecules in ChEMBL differed substantially from any other structure database for small molecules of biological interest. For example, 81.9% of the molecular structures in ChEMBL have a natural product-likeness score below zero, and less than 3% are marked as "natural

product". Manual inspection confirmed that ChEMBL contains numerous small molecules "not of biological interest" (i.e. not primary or secondary metabolites, food additives, drugs, drug degradation products, etc).

"To speed up computations, we estimated (provably correct) lower bounds of all distances. We performed exact computations only if the distance bound is at most a chosen distance threshold, which we set to 10, unless stated otherwise." I have two comments in regard to this: (i) The proof should be referenced in the methods, and (ii) what is the theoretical and practical speed-up?

(i) The two bounds were introduced without weights by Raymond *et al.* (2002), where proofs are given. Including weights is a straight-forward modification, which does not change the validity of the proofs. *We have added a citation to Raymond et al. for the proofs.*

(ii) Computing the bounds requires polynomial (cubic) time, whereas computing the MCES instance via the ILP requires exponential time, in the worst case. *We have added a sentence to the manuscript.* In application, computing bounds is even faster: It boils down to finding a maximum weight perfect matching, and this can often be found in "almost linear time", as augmenting paths are often short in practice.

"t-SNE and Minimum Spanning Trees visualizations are provided as examples in Supplementary Fig. 13." A reference (or a description of the implementation) for the MST method is missing.

Minimum Spanning Tree is a classical problem from theoretical computer science. We use the implementation from SciPy; *we added to the manuscript that it is based on Kruskal's algorithm.*

"To avoid both proliferating running times and cluttered plots, we subsampled 20,000 biomolecular structures" The method for subsampling should be described in detail.

We added "we uniformly subsampled"; further details can be found in the section *Subsampling molecular structures*, see also below.

Fig. 1. Map of biomolecular structures with color-coded compound classes. For comparison, an embedding of a fingerprint space should be included, as the introduction highlights this as an alternative approach. As metabolites are the focus, using the MAP4 fingerprint may be prudent (<https://pubmed.ncbi.nlm.nih.gov/33431010/>).

If we are not mistaken, then this is shown in Supplementary Fig. 15. *We have added the plot for MAP4, which indeed looks somewhat better than the other two.* Yet, we found many examples where the MAP4 Tanimoto behaves counterintuitively, see Supplementary Fig. 14. From the distribution of Tanimoto coefficients for our dataset, MAP4 coefficients are generally much smaller than those of other fingerprints, see below. *We have added a sentence that this must not be misinterpreted as a "measure of quality" for fingerprint types.*

Other types of visualizing distances, including spanning trees (TMAP), are shown in Supplementary Fig. 9.

"If we are able to spot non-uniformness in the 2-dimensional UMAP embedding, then it is presumably not a uniform subsample in higher dimensions, either." Why evaluate the uniformness of the subsample on the UMAP embedding?

Directly spotting non-uniformness in high dimensional data using a small sample is extremely challenging. Reducing the dimension is a classical approach for doing so, compare to Principal Component Analysis. It gets even harder here because molecular structures are not finite-dimensional vectors. We experimented with adding the above sentence but found that it impedes the flow of the paragraph; if the referee insists, we would nevertheless add it.

Distribution of molecular structures in public datasets: "We consider ten public molecular structure datasets frequently used to train machine learning models": It may be important to note, that increasingly often, these sets are "only" used to evaluate models trained on larger data sets (see all the recent transformer-based models) with and without fine-tuning.

Models like GROVER and Chemformer are pre-trained on separate (larger) datasets, and subsequently evaluated on one or more of the MoleculeNet datasets, often with fine-tuning. For these models, the (explicit or implicit) claim is still that they can predict a property of interest. Yet, training and evaluation of the property are executed on the smaller datasets that we investigate in this manuscript. To this end, we would argue that such models might be impaired by the unavailability of training data, as are "regular" end-to-end models. We now mention such models in the manuscript.

"We argue that most of the public datasets are also not representative, meaning that large areas of the biomolecular structures are completely missing in the datasets. In fact, some datasets are concentrated in one or few areas in the plot." As noted in the previous point, many models, or more specifically architectures (which may be another good point to make,

that certain architectures may only work well in specific areas of the chemical space, AFAIK, this has not yet been explored), are merely evaluated on these data sets. It may therefore be interesting to also show the combined coverage of these "moleculenet" sets in an additional plot.

We believe that such a plot may actually be misleading, and would prefer not to include it in the manuscript. See our discussion on Chemformer and GROVER above: A model might be able to learn a better internal representation of the chemical space, but that does not mean that it is able to predict any of the molecular properties for compounds beyond the training data. If the toxicity dataset contains only lipids, then even an excellent internal representation of flavonoids will in no way help us to predict their toxicity.

"Molecular structure into a bit string of fixed length": A binary array (or vector) may be a better description than a "bit string".

Changed to "binary vector".

Furthermore, it should be explained here which method (ECFP?) was used, as the results may differ depending on the chosen method.

We now explicitly mention that the described issues hold for **any** type of molecular fingerprint.

"Using the Jaccard index or the Jaccard distance for similarities and dissimilarities": The connection between Jaccard distance and the Tanimoto coefficient should be discussed.

The Jaccard distance is simply one minus the Jaccard index, hence, one minus the Tanimoto coefficient. We added that to the manuscript. Tanimoto himself introduced a "Tanimoto distance" that is mathematically not elegant (not a metric) and rarely used.

"For completeness, we have also computed an UMAP embedding using distances computed from Tanimoto coefficients". As noted previously, the chosen fingerprint plays a much bigger role than the chosen distance function (Jaccard, in this case).

Supplementary Fig. 14 contains UMAP plots for MACCS, ECFP4/Morgan, and now also MAP4 fingerprints.

Maximum Common Edge Subgraph computations: "Our method is the first to use an ILP for the comparison of chemical structures and, to the best of our knowledge, is also the first public publicly available implementation for exact MCES computations." I am also not aware of an existing implementation with this approach.

There is one now: On Nov 8, 2023, Dave Cosgrove added RascalMCES to RDKit. See <https://greglandrum.github.io/rdkit-blog/posts/2023-11-08-introducingrascalmc.es.html>. We added a short evaluation to the manuscript.

- "For the ILP, 24 of 20,000 instances did not finish within four days of wall clock time." It would be interesting to show these structures in the SI.

Excellent suggestion, see Supplementary Fig. 12.

"To exclude bias through subsampling, we repeated the above analysis using all pairs from the 19,994 biomolecular structures." Where does this number (19,994) come from? By reading the manuscript, I have not encountered it so far.

This is explained in Section *Subsampling molecular structures*, see below; we have added a reference.

It should also be mentioned, that recent model evaluations on these data sets use what is known as Murcko scaffold splits. These account, at least partially, for the issues mentioned, as performance impacts of scaffolds that are not part of the training set are accounted for.

We have added a paragraph on scaffold splits.

Natural product-likeness score distributions: This is a valuable analysis that may even be extended to other properties in a future publication.

Thank you, good idea.

"We consider only "two-dimensional" molecular structures": "molecular topology" may be a better wording, as two-dimensional structure implies an embedding.

We removed the phrase "two-dimensional", although we have seen it being used for this purpose in other chemistry publications. Yet, "topology" hints towards 3D structure, doesn't it? Wikipedia might not be the best source, but it is pretty clear on that: "In chemistry, topology provides a way of describing and predicting the molecular structure within the constraints of three-dimensional (3-D) space."

Biomolecular structures: "There are clearly larger databases such as PubChem, but those databases also contain molecular structures not of biological interest." As mentioned previously, ChEMBL would contain only molecules of biological interest.

As discussed above, one has to be careful with ChEMBL. It is definitely not only "molecules of biological interest".

Subsampling molecular structures, "From this set, we uniformly sampled a subset of cardinality 20,000": As previously noted in the main part, the nature of the uniform sampling is not clear (at least to me). In respect to which property of the molecules was the sampling uniform?

We added "Each of the 718,097 molecular structures has exactly the same probability to be drawn; similarly, each subset of cardinality 20,000 has exactly the same probability to be drawn."

"We later noticed that six of these 20,000 molecular structures are single ions, for example, a single iron ion." I noted previously that it would be good to know where the number "19,994" comes from. As the same will probably happen to other readers, there should be a reference to this methods subsection in the main text or a short explanation should be added to the main.

Done.

Computing the Maximum Weight Common Edge Subgraph: I also have no specific comments on this section. However, I feel like the whole section, with the exception of the detailed description of the ILP, should be in the Results section. On the other hand, the section is fairly technical. So I leave this for the authors to consider.

It is indeed rather technical, so we would leave the decision to the editor.

Conclusion: I agree with this conclusion.

Thank you.

Reviewer #1 (Remarks to the Author):

I am glad to say see that many changes have been made. I would also add it never a good idea to insult the reviewer by assuming that I did not know the Box quote.

Thank you. We did not want to insult anyone, so, sorry for that.

The main issue I still have is with the definitions of 'biomolecular structures of interest' or 'molecules of no particular biological interest': this is defined by the datasets employed, and not by any formal definition such as showing efficacy in man, or found as a natural product, or showing in vitro activity above a threshold. As such, bias has already been introduced, and this should be acknowledged. The whole Pharma business is based on "synthetic molecules"! It is known that halogens are uncommon in natural products, for example, but may be rescued by the NIST or solubility datasets. There are some pointers here <https://www.peter-ertl.com/publications.html> for example reprint 138.

We fully agree that different applications require different molecular domains. Our previous changes to the manuscript, where we tried to explain this, were not for the best. **We have added a short discussion of the issue to the definition of "molecules of biological interest".** With our definition, we are basically sticking to what fields such as (untargeted) metabolomics, environmental research, natural products research, foodomics etc consider to be "molecules of biological interest". A molecule that has been tested for binding as part of a combinatorial library, is usually not found "in the wild", neither in human samples, nor in biological or environmental samples. We are aware that this is not a clean separation, but we do not think that we can ever do better than that. Notably, 16.7% of the molecular structures in our "biomolecule" database contain halogens; **we have added this information to the manuscript.**

For ChEMBL, **we performed a comparison of ChEMBL structures vs. "molecules of biological interest" structures** (as defined above), and **added Supplementary Fig. 16 where we created a unified UMAP embedding for both datasets, then compared their distribution.** Interestingly, the general layout of the joint UMAP plot seems to be dominated by the molecules of biological interest. An explanation could be that natural products and metabolites have a higher scaffold diversity than synthetic molecules, but we don't want to go too far out on a limb, potentially overinterpreting the UMAP plot.

Next, **we repeated our main analysis of embedding nine training datasets into the molecules of biological interest, using ChEMBL as the "background distribution".** See the new **Supplementary Fig. 17.** Notably, the UMAP plot shows a clear separation of compounds that contain a sulfonyl group (upper-left cluster) vs. those that do not. **We briefly discuss the two datasets (Lipo, SMRT) where we observe a notably different coverage. We have also added a paragraph on ChEMBL to the conclusion of the manuscript.**

The yaml file should also include the python version required, if a minimum is required

Done.

Reviewer #3 (Remarks to the Author):

The authors describe an analysis of molecular datasets. They visualize the chemical space of these datasets by computing a UMAP of distances constructed from a "myopic MCES"

metric. The method is interesting and there is potential here, but ultimately what is shown is that different molecular datasets are drawn from different distributions, which isn't particularly surprising or noteworthy (histogramming molecular weights would reach the same conclusion in most cases).

Thank you. We disagree that this is all that the manuscript is about. Machine learning differentiates whether a dataset is *uniform* (that is, all sample points are uniformly drawn from the underlying, unknown distribution) or “*trainable*”. The latter does not require that all parts of the underlying distribution are sampled uniformly; instead, it is sufficient if the dataset carries the necessary information so that a model may generalize for the complete distribution. Our plots are a strong indication that some of the MoleculeNet datasets are not even trainable subsets, at least not for the set of biomolecules that we have chosen to analyze.

With regards to distributions, we suggested using NP-likeness scores in the manuscript, too (Fig. 5). If a dataset differs substantially from the distribution of molecules where we want to apply our model, this should indeed “raise an eyebrow”. Yet, unless differences are as substantial as those shown for certain datasets in Fig. 5, then a raised eyebrow (a note in the manuscript text) may be enough. For example, we have not encountered a small molecule dataset for which the distribution of molecular weights fully agrees with that of the underlying “biomolecules”, see the figure below. But that is not required, as long as molecules of all weights are part of the training data.

The two plots show the distribution of molecular weights, for the different datasets. We can add this figure to the supplementary material and add a sentence to our discussion about the NP-likeness score, at the editor's discretion.

This could be a much more interesting and useful paper if the authors used their method to illustrate the extent various splitting methods (e.g. scaffold split, fingerprint similarity split) are (or are not) effective at constructing truly independent test sets that are an effective measure of the generalizability of trained models.

We have added Supplementary Fig. 18 und 19, which show scaffold splits of nine training datasets and, in more detail, a ten-fold scaffold split of the BBBP dataset. We have also added a brief discussion in the main text. We see this as a demonstration that this type of analysis can indeed be executed with the methods presented in this manuscript. But we

want to stress that our manuscript already has a topic. It deals with the fact that available small molecule datasets are often somewhat restricted, also resulting in a restricted domain of applicability of the resulting models. And, we present a method to detect this. To this end, we would like to keep the split analysis to these figures, so as not to distract too much from the topic of this manuscript. If our method is later used to analyze scaffold splits in detail, this would be excellent.

I do not understand the authors' resistance to including ChEMBL. If a compound is active in a biological assay it is clearly biologically relevant. The fact that ChEMBL compounds apparently have a different distribution from the other molecules considered only strengthens the claim that the space of biologically relevant molecules is much larger than commonly used training sets.

As noted above (R1), we fully agree that different applications require different domains. Our previous changes to the manuscript were not for the best. With our selection of databases, we merely capture what the fields of metabolomics, environmental research and natural products consider as "molecules of biological interest". See the answer to (R1) for details. We have added a short discussion on "biomolecules" to the paragraph introducing "molecules of biological interest", and we have added Supplementary Figures 16 and 17 where we repeat our analysis of coverage for the ChEMBL database.

The authors lament that end-to-end models neither consider or analyze their domain of applicability, yet do not give best practices on how to do that. There seems to be an assumption that only chemical similarity can be used to define a domain of applicability, but this isn't true. A stronger paper might select a state-of-the-art property prediction model that includes confidence scores and map how those scores relate to distance from the training set in UMAP space or demonstrate a relationship between error and distances in MCES distance from the training set.

That would be an interesting paper, but it would also be a different, later paper. We would be happy to read it, but it goes far beyond the topic of this manuscript.

Overall, the methodology is sound and has potential, but the results are not particularly noteworthy.

Thank you. We may disagree on the "particularly noteworthy" part, but then, this clearly lies in the eye of the beholder.

Reviewer #3 (Remarks on code availability):

I scrolled through it - it is well document and looks usable, although I did not try it myself.

Ok.